



# The surface energy balance during foehn events at Joyce Glacier, McMurdo Dry Valleys, Antarctica

Marte G Hofsteenge[1,2], Nicolas J Cullen[1], Carleen H Reijmer[2], Michiel van den Broeke[2], Marwan Katurji[3], and John F Orwin[4,5]

[1]School of Geography, University of Otago, Dunedin, New Zealand
[2]Institute for Marine and Atmospheric Research, Utrecht University, Utrecht, The Netherlands
[3]School of Earth and Environment, University of Canterbury, Christchurch, New Zealand
[4]Resource Stewardship Division, Alberta Environment and Parks, Canada
[5]Department of Geography and Planning, Queen's University, Canada

**Correspondence:** Marte Gé Hofsteenge (marte.hofsteenge@postgrad.otago.ac.nz)

**Abstract.** The McMurdo Dry Valleys (MDV) are a polar desert, where glacial melt is the main source of water to streams and the ecosystem. Summer air temperatures are typically close to zero and therefore foehn events can have a large impact on the meltwater production. A 14-month record of automatic weather station (AWS) data on the Joyce Glacier is used to force a 1D surface energy balance model to study the impact of foehn events on the energy balance. AWS data and 1.7 km resolution

output of the Antarctic Mesoscale Prediction System (AMPS) are used to detect foehn events at the AWS site. Foehn events at Joyce Glacier occur under the presence of cyclones over the Ross Sea. The location of Joyce Glacier on the leeward side of the Royal Society Range during these synoptic events cause foehn warming through isentropic drawdown. This mechanism differs from the foehn warming through gap-flow that was earlier found for other regions in the MDV and highlights the complex interaction of synoptic flow with local topography of the MDV. Shortwave radiation is the primary control on melt at Joyce

Glacier and melt often occurs with subzero air temperatures. During foehn events, melt occurs more frequently and melt rates are enhanced, contributing to 19% of the total annual melt. Foehn winds cause a switch from a diurnal stability regime in the atmospheric surface layer to a continuous energy input from sensible heat flux throughout the day. The sensible heating during foehn is largely compensated for by extra heat losses through sublimation, and melt rates are enhanced through an additional energy surplus from a reduced albedo.

## 1 Introduction

Changes in glacier volume of large polar ice caps impact the global surface energy balance (SEB) and contribute to global sea level rise. On the regional and local scale, changes in glaciers can have a large impact on ecosystems. A clear example of a region where glacial meltwater is vital for the ecosystem are the McMurdo Dry Valleys (MDV), Antarctica. The MDV are a polar desert, where the annual precipitation is less than 50 mm/yr (Fountain et al., 2010). The microbial ecosystem that is

hosted in this extreme climate is one of earth's rarest ecosystems. In the dry and windy climate, most of the snowfall does not accumulate in the valleys but sublimates and glacial meltwater is therefore the main source of water to streams and the ecosys-





tem (Fountain et al., 2010; Eveland et al., 2012). Glacial runoff also plays an important role in nutrient delivery (Dubnick et al., 2017) and biogeochemistry of the MDV lakes (Foreman et al., 2004). Extreme events can severly alter the light and nutrient availability in the lakes (Fountain et al., 2016). Episodic warming events are therefore expected to have long-term effects on the

valley and lake ecosystems (Foreman et al., 2004; Fountain et al., 2016). Improving our understanding of the glacial response to warming events is therefore essential in order to understand how glacier and ecosystem response might change in a warming wold.

The strongest episodic warming events in the MDV are observed in the winter months, which are characterized by long
periods of stable stratification where cold-air pools form at the surface. The pools in Wright Valley can last up to 2 weeks, until they are eroded by strong wind intrusions into the valleys, resulting in a rise in near surface temperatures by up to 25-50 °C (Speirs et al., 2010; Zawar-Reza et al., 2013). The strong down-valley winds in the MDV have previously been explained as strong adiabatically warmed katabatic winds (Bull, 1966; Doran et al., 2002; Nylen et al., 2004). However, more recent research shows that these winds have a foehn warming signature (Speirs et al., 2010, 2013; Steinhoff et al., 2013, 2014). Speirs
et al. (2010) relate the valley wind events to synoptic cyclones over the Ross Sea. The southerly flow branch from the Ross Sea cyclones is initially channeled along the Trans-Antarctic mountain ranges and then forced through high elevation mountain gaps towards the MDV, accelerated by mountain wave activity and brought into the MDV through pressure driven topographic channeling (Steinhoff et al., 2013, 2014).

While foehn events occur more frequently in winter, they can have a larger impact on glacial melt and stream-flow in summer when air temperatures are typically close to 0 °C (Steinhoff et al., 2013). Doran et al. (2008) compared a normal stream flow intensity season with the 2001/02 flood year and linked the increased glacier mass loss and stream-flow to an enhanced frequency of down-valley winds. This relation between the frequency of down-valley wind events and stream-flow also possibly explains the seemingly contradictory presence of large glacial lakes in the MDV during the colder climate of the Last Glacial
Maximum (Doran et al., 2008; Obryk et al., 2017).

Foehn events in the MDV have been associated with enhanced meltwater generation, but no studies have shown how the energy balance components of MDV glaciers are altered during foehn events and what their role is in melt generation. Other Antarctic regions, like the Antarctic Peninsula region that experience more melt compared to the MDV, have received more
attention on the link between foehn and surface melt. The westerly airflow that causes foehn winds crossing the Antarctic Peninsula mountain range reduces cloud cover and increases incoming solar radiation and sensible heating over the Larsen B and C ice shelves (Kuipers Munneke et al., 2018; Elvidge et al., 2020; Turton et al., 2018, 2020; Laffin et al., 2021). This increase in energy at the surface is largely compensated for by a corresponding decrease in incoming longwave radiation and increased latent heat losses (King et al., 2017; Turton et al., 2020). While these studies have demonstrated that shortwave radi-
ation through cloud-clearance has the largest impact on foehn-induced melt, Elvidge et al. (2020) found that most foehn-driven melt occurs under conditions where sensible heating dominates the SEB. Closer to the MDV, 40% of melt of the Ross Ice



Shelf occurs during foehn events (Zou et al., 2021b). The foehn effect increases sensible heating and its associated mesoscale dynamics reduces cloud-cover resulting in extensive downwelling solar radiation over the Ross Ice Shelf, where net solar radiation is the leading component in the SEB during melt (Zou et al., 2021b, a).


The role of foehn winds in melt generation is expected to be different in the colder and drier environment of the MDV compared to the Antarctic Peninsula and Ross Ice Shelf, where melt is more common and abundant. In the MDV, the summertime near-surface air temperatures usually hover around 0 °C (Katurji et al., 2013) and foehn winds can significantly increase air temperatures, bringing the glacier surface temperatures more often to the melting point through sensible heating. At the same

time, the dry and windy foehn conditions are expected to also increase turbulent latent heat losses, cooling the surface and therefore prohibiting melt during these events but enhancing sublimation.

The research aim of this paper is to describe the dominant energy fluxes during foehn and non-foehn conditions in order to resolve the role of foehn occurrence in meltwater generation for a MDV glacier. We use a 14-month record of automatic

weather station (AWS) observations on the Joyce Glacier to drive a 1D SEB model. The Joyce Glacier is located south-east of other MDVs separated by the Royal Society Range and has received little attention in foehn and energy balance studies. The AWS observations in combination with high-resolution atmospheric model output of the Antarctic Mesoscale Prediction System (AMPS) are used to detect foehn events at Joyce Glacier.

## 2 Methods and data

In this study, we use a 14-month record of observations from an automatic weather station (AWS) located at the Joyce Glacier to force a SEB model (Figure 1). This continuous data-set covers two melt seasons and one winter season, from November 2008 to January 2010. Foehn events are detected from local AWS observations and spatial information from model output of AMPS. The AMPS output is also used to describe the spatial signature of foehn events at Joyce Glacier.


### 2.1 AWS data

The AWS was installed on Joyce Glacier at 78°01.245'S, 163°47.608'E, roughly 400 m above sea level on 26 November 2008 and removed on 31 January 2010. The glacier tongue is located at the western end of the Garwood Valley, a coastal valley east of the Royal Society Range (about 25 km from McMurdo Sound). The station was set-up in the ablation zone of the

Joyce Glacier, about 150 m from the northeast-cliff. The glacier tongue is approximately 2 km wide and the equilibrium line is roughly 3.5 km up from the glacier terminus at an elevation of 600 m.




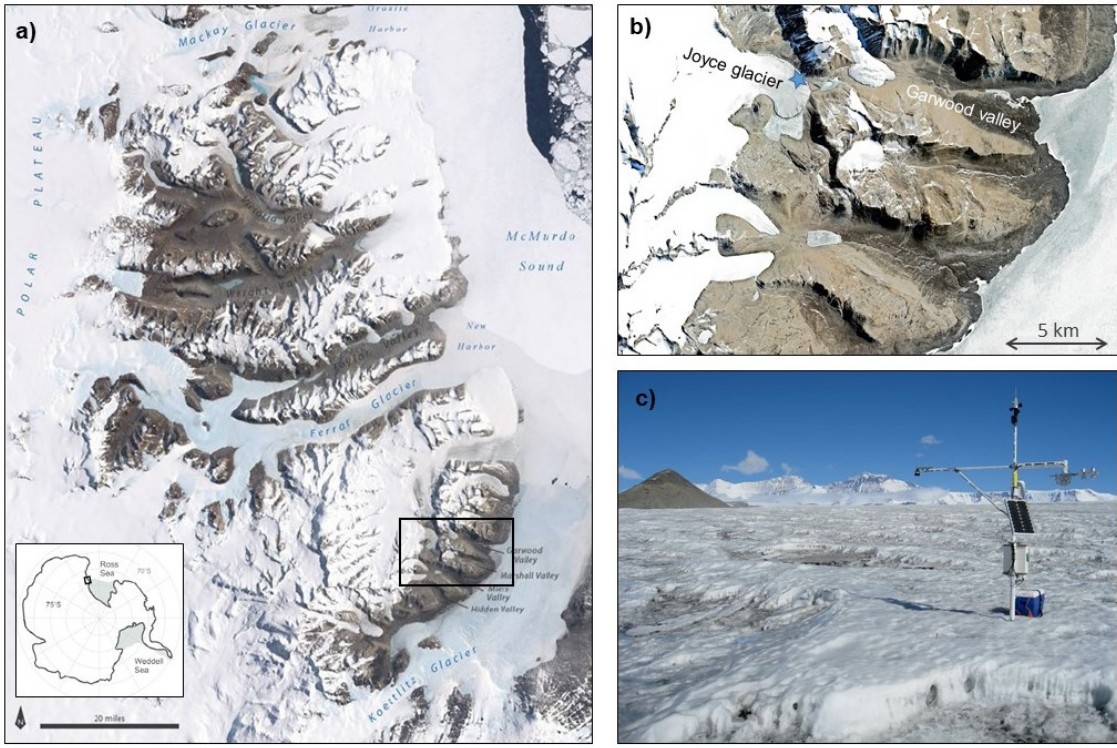

**Figure 1. a)** Map of the McMurdo Dry Valleys (Hogg and Wall, 2012) where the black box shows the region with Garwood Valley and Joyce Glacier shown in **b)**. **c)** AWS set-up on Joyce Glacier. Blue star in **b)** marks location of the AWS.

Continuous measurements at 15 minute resolution of air temperature, relative humidity, incoming and outgoing longwave and shortwave radiation, wind speed and direction and air pressure were obtained. Additionally, the distance to the ice surface

was measured using a sonic ranger on the mast for instrument height, ablation and snowfall estimation. The instruments that were used are listed in Appendix A, Table A1 and were the same as those described in Gillett and Cullen (2011). The measurements were taken every 300 seconds and stored as 15-minute averages on a data logger.

Data processing was similar to that described in Gillett and Cullen (2011), and includes recalculation of relative humidity

measurements to correct for saturation with respect to ice instead of liquid water (Steffen and Box, 2001). Accumulated albedo was calculated from accumulated incoming and outgoing shortwave radiation over 24 hours as in Van Den Broeke et al. (2004) and filtered for outliers by removing data outside the 2nd and 98th percentile. The surface height change record was cleaned by manually removing spikes from the data and applying a 2-hour running mean through the data. A snow accumulation record was calculated from daily differences in surface height, using a lower threshold of 1 cm for precipitation to occur.



## 2.2 Energy and mass balance model

We used the energy balance model presented by Reijmer and Oerlemans (2002) and further developed as described in Jakobs et al. (2019), which assumes the energy balance for a surface skin layer by solving the following equation for skin temperature:

$$Q_M = S_{in}(1-\alpha) + Q_P + L_{in} + L_{out} + SH + LH + Q_G, \tag{1}$$

where $Q_M$ is the energy used for melt, which is 0 when the surface temperature $T_s < 273.15$ K, $S_{in}$ the incoming shortwave radiation, $\alpha$ the surface albedo, $Q_P$ the amount of penetrated shortwave radiation, $L_{in}$ and $L_{out}$ the incoming and outgoing longwave radiation, $SH$ the turbulent heat flux, $LH$ the latent heat flux and $Q_G$ is the sub-surface conductive energy flux. All fluxes directed towards the surface are defined as positive. Energy supplied by rainfall is not relevant for the MDV glaciers and therefore is neglected.

$S_{in}$ is calculated from observed $S_{out}$ and accumulated albedo and $L_{in}$ is based on AWS observations. Outgoing longwave radiation is calculated using Stefan Boltzmann's law from the model calculated skin layer temperature. The turbulent heat fluxes $SH$ and $LH$ are calculated with the bulk aerodynamic method, considering atmospheric stability based on the Monin-Obukov similarity theory. Measurements of temperature, wind speed and temperature at the AWS measuring level are used together with the calculated skin surface temperature, zero wind speed and saturated humidity at the surface to calculate the turbulent heat fluxes.

Solar radiation is able to penetrate the snowpack and attenuation of the shortwave radiation is calculated separately for 118 different wavelength bands using the model of Brandt and Warren (1993), also used by Van Den Broeke et al. (2008) and Kuipers Munneke et al. (2009). The solar radiation that is able to penetrate the snowpack $Q_P$, is removed from the surface layer energy balance and added to the appropriate subsurface model layers. The parameter *dzrad*, the surface layer thickness used in solar radiation penetration calculation, determines this partitioning of the penetrated radiation between the surface layer and subsurface energy budgets. An ice grain radius of 2.8 mm is used for the solar radiation penetration routine, as was measured for Taylor glacier by Hoffman et al. (2008).

The subsurface model is based on the SOMARS model (Reijmer and Hock, 2008). The vertical subsurface grid is composed of layers with thickness varying with increasing layer thickness with depth. The parameter *dz0* sets the thickness of the upper grid-cell. Internal melt occurs when the subsurface temperatures of a layer reach 0 °C and the excess energy is used for melt. Internal meltwater is added to the water content of that layer and can drain to deeper layers and refreeze if temperatures are below 0 °C. Refreezing in turn increases the subsurface temperature and density. Subsurface meltwater that percolates down to the firn-ice interface or subsurface melt that occurs in the ice runs off. To give an estimation of the uncertainty involved in parameter choices, we ran the model as an ensemble using different parameter combinations. Table 1 shows the four parameters that are considered and the set of values that is used in the ensemble.





**Table 1.** Overview of parameter settings used in ensemble simulations with the surface energy and mass balance model. Parameter settings of the reference run are indicated in bold.

|  | Parameter name | Parameter values |
| --- | --- | --- |
| dzrad | surface layer thickness for radiation penetration | 0, 0.05, 0.01, **0.005**, 0.001 [m] |
| z0 | aerodynamic roughness length for momentum | 0.0001, 0.0005, 0.001, 0.005, 0.01, **0.05** [m] |
| dz0 | thickness upper grid-cell | 0.02, **0.03**, 0.04, 0.05, 0.1 [m] |

## 2.3 Antarctic Mesoscale Prediction System

The numerical weather prediction system AMPS is used as a real-time operational forecasting tool for Antarctica by the National Center for Atmospheric Research (NCAR), USA. AMPS is based on the polar version of the Weather Research and Forcasting (WRF) model. The boundary and initial conditions of AMPS simulations are taken from output of the Global Forecasting System (GFS) and the model is initialized twice daily. The first 12 h after initialisation is treated as model spin-up and disregarded.

AMPS is run on 5 or 6 domains over Antarctica and its subregions and archived for different domains and resolutions. In this study we use output of AMPS WRF 45 version for the year 2009 for domain 5. This domain is centered over Ross Island and covers the McMurdo Dry Valleys with a horizontal spatial resolution of 1.67 km. AMPS is run on 44 model levels, and this study uses AMPS products that were reduced to the first 15 model levels and 6 fixed pressure levels. In addition, the largest domain is used for synoptic scale analysis, covering the Antarctic continent with a 15 km horizontal resolution.

The AMPS output has been used previously in Antarctic foehn and other meteorological studies (Speirs et al., 2010, 2013; King et al., 2017; Katurji et al., 2019; Turton et al., 2018, 2020). Speirs et al. (2010) evaluated high resolution AMPS output with AWS observations in the MDV and concluded that the model output is suitable to study regional airflow and synoptic conditions during foehn in the MDVs. While AMPS can have difficulties representing the near-surface conditions on the valley floors, it performed well in simulating the foehn onset and cessation (Speirs et al., 2010).

## 2.4 Foehn detection

In this study, we use three different classification criteria to detect foehn events from AWS and model data and combine them for a final foehn detection algorithm. The evaluation of the performance of the different methods is presented in Sect. 3.1.

### 2.4.1 Speirs

The first detection method was designed for foehn events in the MDV by Speirs et al. (2010) and adapted by Steinhoff et al. (2014). It is based on thresholds in relative humidity, temperature and wind speed and direction. Wind direction is adapted for



local conditions over Joyce Glacier. Since the AMPS simulated wind direction over the Joyce Glacier does not represent the observed wind direction well, a different wind direction criterion is used for the AWS and AMPS data (Sect. 3.1). Using the definition of Steinhoff et al. (2014), all the following conditions must be met over any 6-hourly period to be classified as a foehn event:

1. Temperature: increase of 1°C over a 1 h period OR any measurement above 0°C.

2. Relative humidity: a decrease of 5% over a 1 h period OR any measurement below 30%.

3. Windspeed: higher than 5 m/s for at least 80% of the observations (in this study 5 out of 6 hours).

4. Wind direction: between 67.5 and 180° when using AWS data and between 112.5 and 225° when using AMPS surface data for at least 80% of the observations.

### 2.4.2 Wiesenekker

Wiesenekker et al. (2018) proposed a method for foehn detection at Larsen C ice shelf in the Antarctic Peninsula based on the fact that foehn are strong wind events from a known wind direction. The following conditions must be met over a 6 h period to be classified as foehn:

1. Wind speed: at least 4 m/s.

2. Wind direction: between 67.5 and 180° when using AWS data and between 112.5 and 225° when using AMPS surface data.

### 2.4.3 Turton

Compared to the relatively simple criteria used by Wiesenekker et al. (2018), Turton et al. (2018) developed a more complex method where a combination of criteria to detect surface foehn conditions and foehn characteristics in the vertical atmospheric structure are used for foehn detection over the Antarctic Peninsula. The surface criteria are focused on the drying characteristic of the foehn-winds and therefore use mainly relative humidity thresholds. The conditions are flagged as foehn when both the specific surface conditions in the observations are met, as well as 'isentropic drawdown' is detected in the AMPS output. Isentropic drawdown is the drawdown of the potentially warmer and drier air from higher elevations when airflow is forced over a mountain range and down the leeward side. This mechanism was earlier used to explain foehn winds at the Antarctic Peninsula (Elvidge and Renfrew, 2016).

A combination of the following two criteria must hold over a 6 h period for it to be flagged as a foehn event:

1. Surface characteristics, one of the following criteria must hold throughout the 6 h period:

    – Fall within a 12 h period with a decrease of 15% in relative humidity.





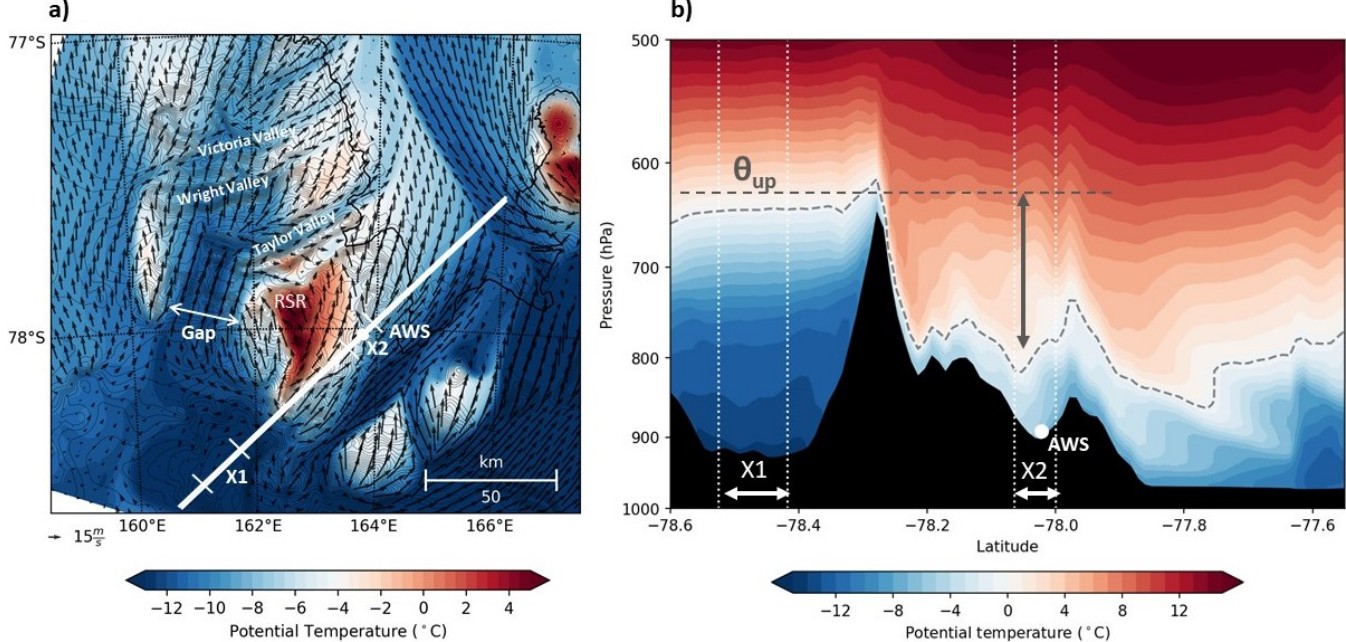

**Figure 2.** Foehn event on the 12th of April 2009 with in **a)** potential temperature at lowest model level, arrows show the 10 m winds and RSR indicates the Royal Society Range and **b)** vertical potential temperature profile across the transect shown in **a**. X1 indicates the upwind region of undisturbed flow, X2 is the region over Joyce Glacier, and $\theta_{up}$ the potential temperature at mountain height level over X1 that is used to measure the isentropic drawdown.

– Average relative humidity record below the 10th percentile value.

– Fall within a 12 h period with a temperature increase of at least 3°C in combination with an average relative humidity record below the 15th percentile.

190    2. Sufficient isentropic drawdown, determined as follows (Figure 2):

– Determine $\theta_{up}$, the average potential temperature and atmospheric pressure over X1 at mountain top height.

– Determine the minimum pressure height of $\theta_{up}$ over X2.

– The pressure height difference between the mountain top height with the $\theta_{up}$ isentrope height at X2 must be at least 50 hPa.

195    All methods are tested using hourly AWS and AMPS data.



### 2.4.4 Foehn detection method used in this study

In this study we combine the above methods using AWS meteorological observations and identify a period as a foehn event when at least 2 out of the 3 methods detected foehn conditions within a 6 h period. Using the agreement of 2 out of 3 method limits the false detection of foehn-events by any single method. This combined method also reduces the sensitivity to the chosen thresholds values of the detection criteria of the individual methods.

## 3 Foehn detection and model evaluation

### 3.1 Evaluation of foehn detection methods

We applied the 3 methods described in Sect. 2.4 to identify foehn conditions at the Joyce Glacier in both AWS and AMPS data for the 14-month observational period. Figure 3 shows an example of foehn conditions at Joyce Glacier in April 2009 based on AWS and AMPS output. The model point chosen here was the closest ice point in terms of location to the Joyce AWS. The temperature based on AMPS was corrected for the 150 m elevation difference between grid-cell and AWS elevation, by using a monthly average temperature lapse rate calculated from the surrounding model points. A clear difference is visible between the observed and AMPS simulated wind direction (Fig. 3a). The AWS was positioned relatively close to the valley sides and the deviation in the wind direction through topographic effects is likely not captured in AMPS. The general wind-regime agrees well between model and observation: both show a dominant wind direction with low wind speeds that changes abruptly to a different direction with stronger winds. On average the strong wind events tend to have higher wind speeds in the AMPS model compared to observations. In contrast, the wind speeds during calm conditions are underestimated in the AMPS model, suggesting a poor simulation of down-glacier winds. While differences are visible in the surface conditions between AMPS and observations, the model performs well in simulating foehn onset and cessation (Figure 3).

The Speirs method uses criteria on sudden changes in relative humidity and temperature and therefore detects mainly the onset of the foehn period (Figure 3). Foehn periods detected by the Speirs method are on average shorter compared to the other methods and do not cover the full period where the typical windy, dry and warm conditions last. There is little agreement in the Speirs detection method between AWS and AMPS data. In the AWS data, 488 hours were flagged with foehn conditions, and 254 in the AMPS data, of which 106 hours agreed with the AWS foehn hours (Appendix B, Figure B1). The difference between the AMPS and AWS data can be related to the fact that changes in the simulated temperature and humidity are typically slower than observed, indicating that other thresholds might be needed to use the Speirs method on model data. Adjusting the threshold to changes over a 2-hour to instead of one-hour period, results in almost 200 more hours detected as foehn in the AMPS data and more than doubles the hours that agree between the AMPS and AWS data.

The Wiesenekker method gives the longest foehn periods, as this method only uses periods of strong winds from one direction, and is not limited by temperature and humidity criteria. In the AMPS data, 13 additional foehn events are detected by the





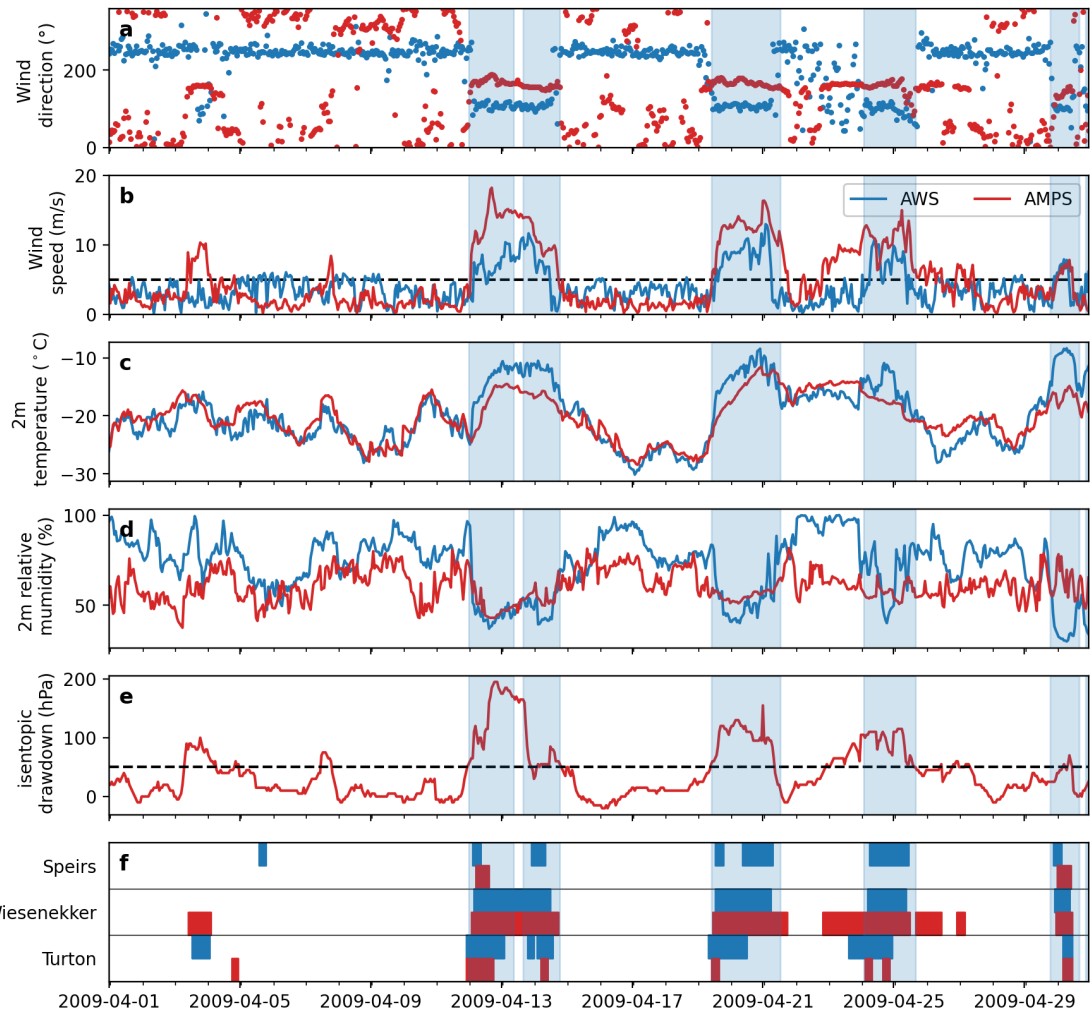

**Figure 3. a-d)** Observed and AMPS simulated wind direction, wind speed, temperature and relative humidity at the Joyce Glacier for April 2009. **e)** Isentropic drawdown calculated from AMPS and **f)** the periods where foehn was detected in each of the methods using surface conditions from AWS (blue) and AMPS (red). Blue shading indicates when foehn was flagged combining at least 2 different detection methods in AWS data. Note that wind speed is given at 10 m for AMPS data and at sensor height for AWS data.

Wiesenekker method. These are short events with stronger winds that were not observed at Joyce, such as for the 4th of April

in Figure 3. Finally, the Turton method gives the largest amount of foehn hours detected on the AWS data (Appendix B, Figure B1). The method gives 934 foehn hours in AWS, compared to 445 on AMPS data. These differences likely follow from the slower and weaker simulated humidity drops in AMPS.





The periods that are flagged as foehn in this study are generally longer than indicated by the individual methods above, but

we are confident they cover the period between foehn onset and cessation (blue shading in Figure 3). The wind speed and wind direction criteria of the Wiesenekker method play a dominant role in the detection and either the Speirs or Turton method confirms the drying and warming characteristic of the foehn event. In total 26 foehn periods occurred during the observation period, counting for 9% of the study period and occurring on 14% of the days. April and November were the months with the most foehn hours, accounting together for almost 40% of the total (Appendix B, Figure B1). During the melt season (December

and January) 6 foehn events occurred, which covered 6% of the period. This lower foehn frequency in summer compared to winter has been found in other MDV studies as well (Nylen et al., 2004; Speirs et al., 2010) and is related to weaker cyclone activity over the Ross Sea region. The mean foehn duration is 37 hours, with a minimum duration of 6 hours and a maximum duration of 90 hours. The highest temperature and lowest humidity of 5.8°C and 23 % were measured during a foehn event on 26 December 2008.


## 3.2   Evaluation of the energy balance model

The energy and mass balance model is evaluated against observed surface height change and surface temperature. The optimal simulation was chosen based on the smallest root-mean-square error (RMSE) and bias between observed and simulated surface temperatures (Figure 4a). The ensemble spread, based on ensemble standard deviation with respect to ensemble mean, is pro-

vided to indicate uncertainty in the simulated surface height change introduced by parameter settings. The optimal simulated surface temperature of the simulation ensemble agrees well with the estimated surface temperature from observed outgoing longwave radiation using the Stefan-Boltzmann law and an emissivity of 1 (RMSE 1.44°C, Figure 4a). The model only slightly overestimates the surface temperature (bias 0.02 °C). In summer, the simulated surface temperature reaches melt (0 °C) more frequently than observed (Figure 4a).


The simulated and observed surface height change agree reasonably well in the December and January months when melt occurs (Figure 4b). During the first melt season (melt1, Figure 4b) 12 cm ice was lost in the observations, compared to 9 cm in the simulation. In the second melt season (melt2), ablation was around 8 cm in both the observations and the simulation. The two snow accumulation events during the melt periods correspond with observations, since the surface height record was used

to determine the accumulation rate.

The largest deviations between simulated and observed ablation are for the period from February until May (subl1, Figure 4b). Simulated ablation is 1 cm, compared to 9 cm in the observations. The largest ablation is observed during the second half of April, during which several foehn events occurred (Appendix B, Figure B1). Sublimation rates calculated by the bulk method

might be underestimated during these foehn events in the simulation. Still, the SR50 surface record should be interpreted with care since the observed surface height changes can be caused by snow redistribution, which is not captured in the simulations. There is better agreement during the May - December period, where the surface height change through sublimation is 5 cm for





both model and observations, although the distribution in time differs.

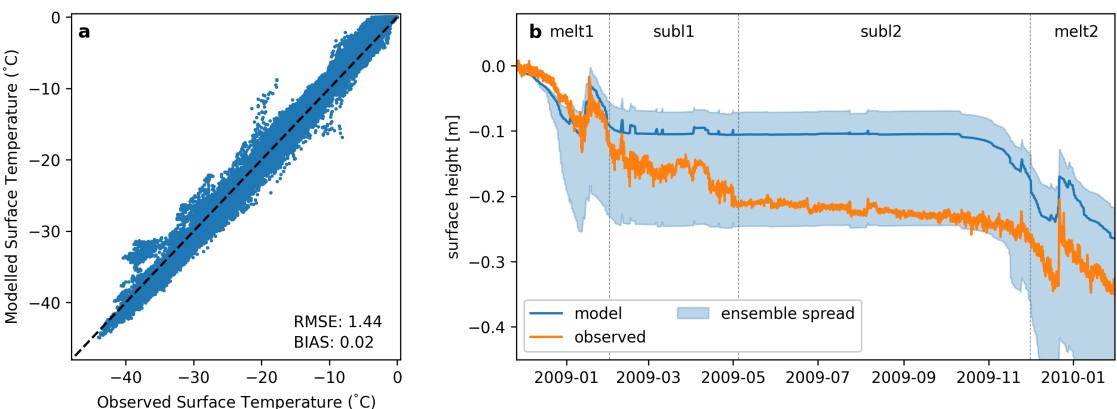

**Figure 4. a)** Observed versus modelled surface temperature and **b)** surface height change, where model represents the optimal ensemble simulation. In **b)**, blue shading indicates the ensemble spread represented by the standard deviation with respect to the ensemble mean.

The ensemble spread indicates the uncertainty introduced by parameter settings in the SEB model (Fig 4b). During the melt seasons the standard deviation in the simulation ensemble increases, showing the uncertainty in the simulated surface height as controlled by uncertainty in simulated melt and sublimation. Over the winter period the standard deviation stays rather constant, indicating that most of the uncertainty in surface ablation is related to the simulation of melt.

The largest uncertainty in the simulated surface height change is due to the *dzrad* parameter, which determines the partitioning of solar-radiation to the subsurface. The grid-resolution of the upper snow layers also largely impacts the simulated surface melt. The importance of solar radiation penetration and the uncertainty that is associated with the partitioning of solar radiation for MDV glaciers are also noted by Hoffman et al. (2014). In this paper we don't focus on the amount of melt that occurs as surface or internal melt, but rather on the occurrence of melt in relation to foehn events. The good agreement between the
observed and simulated surface temperature gives us confidence in using the model to study the dominant energy fluxes and melt occurence under foehn and non-foehn conditions.

## 4 Results

### 4.1 Foehn occurrence at Joyce

A mean sea level pressure anomaly plot during all foehn hours reveals the presence of low pressure systems over the Ross Sea during these events (Figure 5a). The dark blue shading in Figure 5a shows the average location of cyclones during foehn





events at Joyce, rather than individual events. These results are consistent with earlier studies that link foehn events observed in Taylor, Wright and Victoria Valley with cyclones over the Ross Sea (Speirs et al., 2010; Steinhoff et al., 2013, 2014).

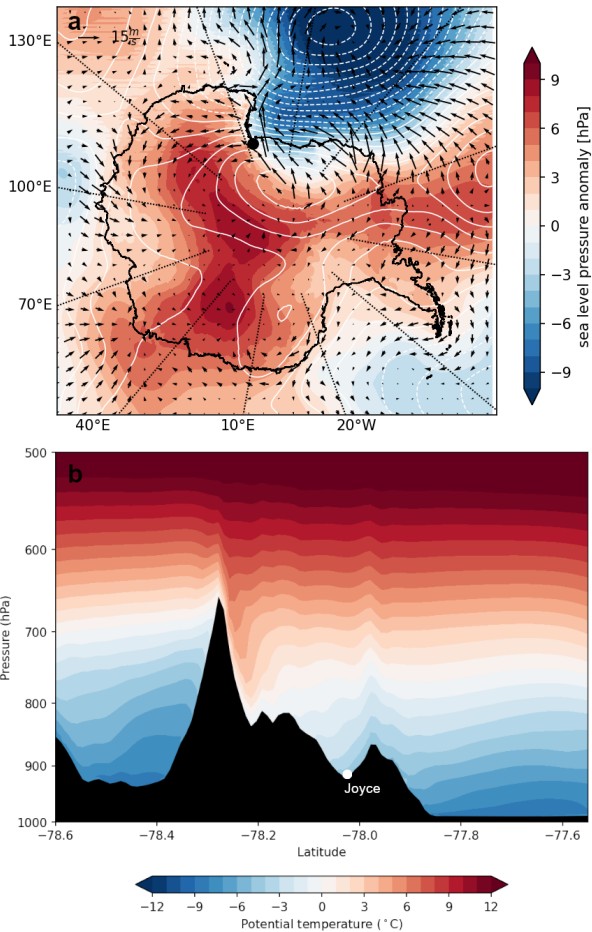

**Figure 5. a)** Sea level pressure (shading), 500 hPa geopotential height (contour) and 10m wind anomaly over the Antarctic domain for foehn events at Joyce Glacier. Foehn anomaly is calculated by subtracting the mean during non-foehn conditions from the mean during foehn events. **b)** Composite of potential temperature profile over the Joyce transect during foehn events at Joyce Glacier.

The presence of the cyclone results in a southerly flow from the Ross Ice Shelf northward towards the MDV (Figure 5a). Foehn events in the Taylor, Wright and Victoria Valley were initiated by similar southerly flows that were forced through the elevated gap between the Royal Society Range and the Antarctic ice sheet (Steinhoff et al., 2013, 2014). The gap is shown for a case in April 2009 in Figure 2a. This gap flow is then accelerated by mountain waves leading to a foehn effect. The heated and dried air was brought to the eastern side of the valleys by pressure-driven channeling (Steinhoff et al., 2013, 2014). This 295    generalized sequence of processes does not hold for foehn conditions at Joyce Glacier because of its location on the eastern





side of the Royal Society Range (Figure 2a).

Part of the observed southerly flow is blocked at low-levels by the Royal Society Range and is split into a component flowing into the gap on the western side of the mountain range, and another component on the eastern side (Figure 2a). This block-
ing and split of the low-level flow by the Royal Society Range was also found for a clustered synoptic circulation pattern of anomalous low sea level pressure over the Ross Sea and resulting strong synoptic southerly flow (Katurji et al., 2019). The low-level blocking results in warmer air from aloft that is drawn down on the leeward side towards Joyce Glacier (Figure 5b). The strongest warming is visible at the base of the north-east slope (Figure 5b). This foehn warming, defined as the increase of temperature in the surface compared to the temperature at the same altitude on the windward side, is on average still about 5
°C over the Joyce Glacier. The April 2009 case in Figure 2b agrees with the average potential temperature field during foehn (Figure 5b) and confirms the isentropic drawdown mechanisms causing the foehn signature at Joyce Glacier.

## 4.2 The energy balance of the Joyce Glacier

The absence of sunlight in austral winter (April - September) ensures that the energy balance is often dominated by longwave
radiative cooling (Figure 6). The average net energy loss by longwave radiation in these months is 47 W/m$^2$, compared to a gain of energy of 40 W/m$^2$ by sensible heating. An average subsurface flux of 6 W/m$^2$ provides energy to the surface and cools the subsurface. The latent heat flux plays only a minor role in the winter energy balance, and switches between slightly positive or negative.

March and October can be considered as transition months with a decreasing (increasing) contribution of solar radiation (Obryk et al., 2020). In these months the longwave radiation remains the dominant cooling component and ablation is limited through the small contribution of the latent heat loss and solar radiation in those months (Figure 6) .

Between November and February, the largest source of energy is solar radiation. Net solar radiation ($S_{in}(1-\alpha)$) is about
111 W/m$^2$, of which 43% penetrates into the subsurface (Q$_P$, Figure 6). The warmed subsurface returns energy to the surface balance through a subsurface conductive heat flux of 42 W/m$^2$. Longwave radiative cooling remains the dominant energy loss, but the latent heat flux plays a larger role in summer, with 28 W/m$^2$ on average used for sublimation. During most of summer the sign of $SH$ is negative, indicating a relatively warm surface close to the melting point and unstable conditions. The albedo of $\approx 0.6$, and large incoming solar radiation heat the subsurface and surface and give rise to this instability in the surface
layer. Although simulated surface temperatures are closer to zero compared to observed surface temperatures (Sect. 3.2) based on both simulated and observed surface temperatures, the sign of the $SH$ is most frequently negative, switching to positive occasionally at nighttime when solar warming is reduced (Sect. 4.4). The unstable conditions are uncommon for ablation zones of most glaciers in summer where the atmosphere is often warmer compared to the surface at the melting point. However, such unstable conditions have previously been found for blue ice areas in Antarctica (Bintanja and van den Broeke, 1994) and dry





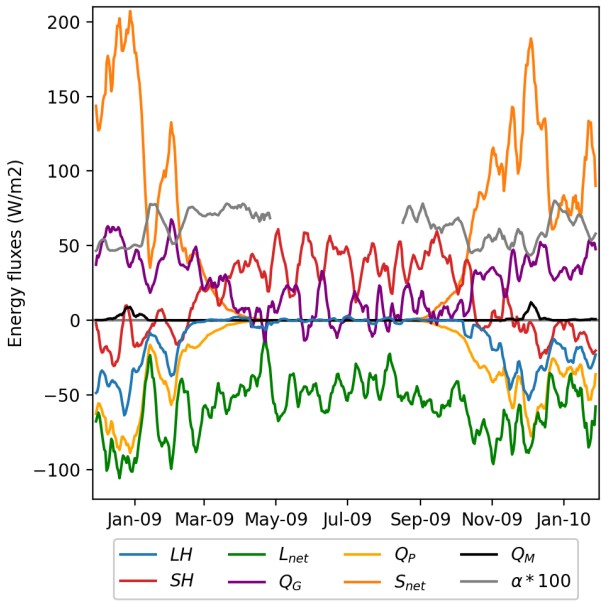

**Figure 6.** Daily averaged energy balance components during the observational record at Joyce Glacier. A 7-day moving average is applied to improve visibility.

snow zones in Greenland and elsewhere (Cullen and Steffen, 2001). The surplus of energy available for melting is small and mostly limited to the December and January months.

### 4.3  Foehn impact on winter energy balance

The energy balance shifts from longwave radiation to sensible heating being dominant in magnitude during the 14 foehn periods in winter. The warm and windy foehn conditions almost double the contribution of sensible heating, to an average flux of 335  72 W/m$^2$. Longwave-equivalent cloudiness ($N_\epsilon$) is estimated from air temperature and incoming longwave radiation using the method of Van Den Broeke et al. (2004), and is on average 0.38 during foehn compared to 0.15 during non-foehn conditions in winter. The cloudier conditions during winter foehn lead to longwave radiative cooling being reduced to 29 W/m$^2$. Figure 7 shows this shift in the energy balance during a number of foehn events in April 2009. During the non-foehn period, radiative cooling is strong under clear-sky conditions and the subsurface cools (Figure 7a). Katabatic winds over the glacier surface still 340  retain sensible heat fluxes up to 70 W/m$^2$ during non-foehn conditions. When the foehn conditions are established, the sensible heat flux increases to values around 100 W/m$^2$ (Figure 7a).

During most of the winter the latent heat flux is slightly positive, resulting in a small amount of deposition. The sign switches when foehn warming is initiated. On average, 9 W/m$^2$ is used for sublimation at the surface during these windy and dry con- 345  ditions. While this pattern is clearly visible in Figure 7 for foehn in April 2009, the magnitude of the sublimation might be





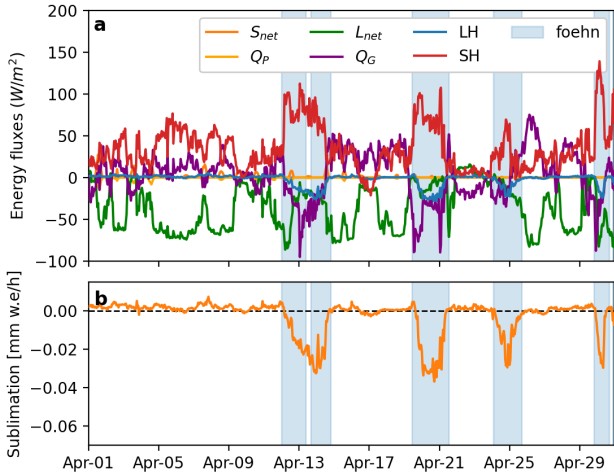

**Figure 7. a)** Energy balance components and **b)** simulated sublimation during April 2009. Background shading indicates periods that are flagged as foehn.

underestimated as was described in Sect. 3.2. The energy losses through sublimation and longwave radiation during winter foehn are only partly compensating the strong sensible heating and therefore the subsurface is also warming through an average flux of -35 W/m$^2$.

### 4.4 Foehn impact on melt

Melt at the Joyce Glacier occurs almost solely in December and January, when net solar radiation is at its peak. Melt at MDV glaciers is driven by solar radiation, and does not occur on days with cloud cover (Appendix C, fig C1). On clear-sky days $S_{net}$ brings the surface temperatures to zero, allowing melt to occur. During the 2008/09 melt season, December and January had an average $S_{net}$ of 136 W/m$^2$, compared to 106 W/m$^2$ over the 2009/10 melt season. This difference was also found for the Commonwealth, Beacon and Taylor glaciers and several valley and lake stations in the MDV (Obryk et al., 2020). The long period of clear sky conditions in 2008/09 and a continuous low albedo of around 0.5 resulted in a large $S_{net}$ and a long period of melt occurring each day (Figure 8). During the entire 2008/09 melt season, only two foehn events close to each other are detected, showing that foehn events are not needed to generate sufficient energy for melt at Joyce Glacier. However, the foehn event appear to enhance melt rates. On 25 December 2008, energy losses from sublimation are large and ablation through sublimation during this foehn event is the highest for the entire study period. Most of December is characterized by unstable conditions, with a surface that cools through $SH$. This changes when the foehn event sets in and the warmer, windier conditions cause a switch in the direction of the sensible heat flux, increasing the energy available for melt. The peak in melt energy during this season occurs just after the foehn event on 29 December 2008. At this point the surface has warmed, $S_{net}$ is still large and cooling through sublimation has decreased. Importantly, the foehn event supplies additional energy to the surface





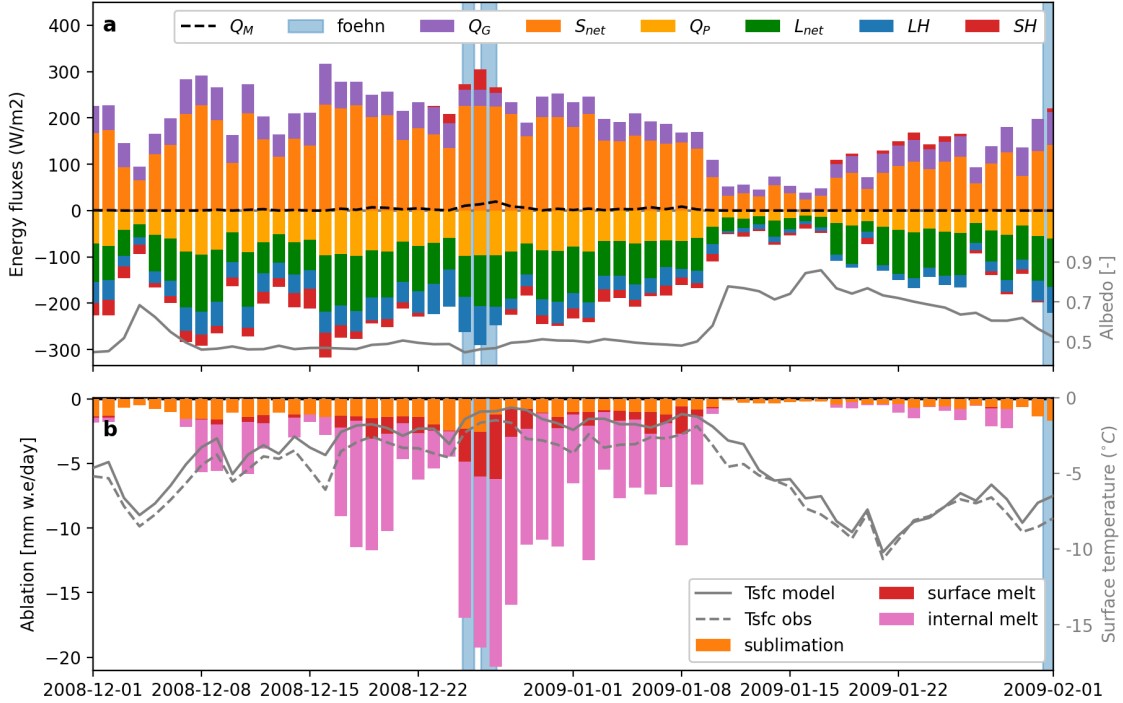

**Figure 8. a)** Daily average energy balance components and **b)** ablation components at Joyce Glacier during the 2008/09 melt season. Grey solid line in **a)** gives albedo and grey solid and dashed line in **b)** represent the simulated and observed mean surface temperature. Blue shading indicates foehn events.

and results in hourly surface temperatures being equal to zero (melting, Figure 8), leading to enhanced melt after the foehn event. Melting continues for another 2 weeks until cloudier conditions and new snowfall reduces the $S_{net}$ and prohibits melt for most of the remaining season.

During the second melt season atmospheric conditions are more variable, with more clouds, and the glacier's albedo increases more frequently through snowfall events or snow redistribution (Figure 9). Net solar radiation is therefore smaller and fewer days result in energy being available for melt (Figure 8). While most melt occurs during December and January in the MDV, a short melt event is simulated in November after a foehn event (Figure 9). The foehn warming causes a rise in the subsurface and surface temperatures. The higher surface temperatures in combination with sunny conditions following the foehn event result in the simulated melt event.

Compared to the 2008/09 season, the 2009/10 has fewer periods of large $S_{net}$, reducing the solar heating of the surface and melt occurrence. Foehn events may play a key role in the melt occurrence. The largest melt rates in this season occur during or just after foehn events (Figure 9). Foehn conditions occur, averaged over both melt seasons, for 6% of the time, but melt during





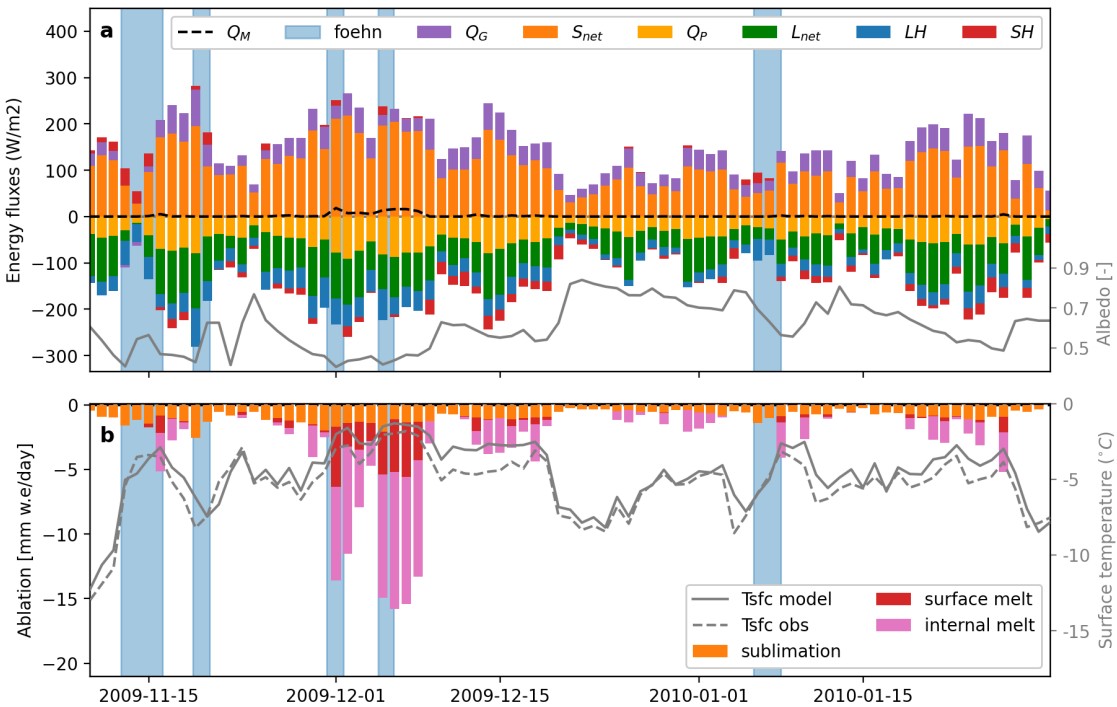

**Figure 9.** As in Figure 8, but for the 2009/10 melt season.

foehn conditions contributes to 19% of the total melt. Melt occurs more frequently under foehn conditions as the foehn events

provide additional energy to the surface. This is visible from the rapidly increasing surface temperatures during foehn events, bringing the surface to the melting point (Figure 9b). The warming of the ice is also caused by the albedo that decreases, for example, during the foehn event on 6 January 2010 as the surface transitions from snow-covered to snow-free (Figure 9a). The pattern of cloudier conditions during foehn events in winter is not evident for the summer foehn events, where the longwave equivalent cloudiness ($N_\epsilon$) is varying and on average 0.33 for both non-foehn and foehn conditions. Atmospheric transmission

($S_{in}/S_{TOA}$) can be used as another indicator for cloudiness and does not reveal any clear pattern in cloudiness during foehn conditions (0.52 and 0.56 during foehn and non-foehn conditions respectively).

Melt occurs not only more frequently under foehn conditions, but the energy available for melt when the surface is at melting point is larger under foehn compared to non-foehn conditions (40 W/m$^2$, compared to 32 W/m$^2$ respectively). The warmer

foehn-air results in a strong increase in the contribution of $SH$ under melt (+33 W/m$^2$), which is largely compensated by an increased energy loss to sublimation (-29 W/m$^2$). The other surplus of energy is provided by an increase in $S_{net}$ as the albedo is on average lower during foehn-events.





## 4.5 Foehn effects on surface layer stability and wind regime

Foehn events change the typical and unique melt conditions of the MDV, where surface melt often occurs with air temperatures below zero. Figure 10 shows the average meteorological conditions during and outside foehn days in the melt seasons used in Figure 8 and 9. On days without foehn-conditions, 68% of the hours the surface melts, the air temperature is below zero. Melt occurs through strong solar heating that brings the surface to melting. The net solar flux is peaking just after mid-day, leading to diurnal variability in the stability of the surface layer over the glacier surface. (Figure 10). The relatively warm surface with

a colder atmosphere above leads to unstable conditions, reflected in a negative $SH$ in Figure 10c. The conditions switch to neutral during the night when solar heating of the surface is reduced. Similarly, a diurnal cycle in atmospheric stability over soil was found in the nearby Miers Valley, induced by topographic shading at night (Katurji et al., 2013). During foehn conditions, air temperatures are more frequently above zero (86% of the time) and $SH$ forms a source of energy to the surface. The distinct diurnal pattern in stability shifts to mostly stable conditions and near-neutral stability at mid-day (Figure 10b, d).


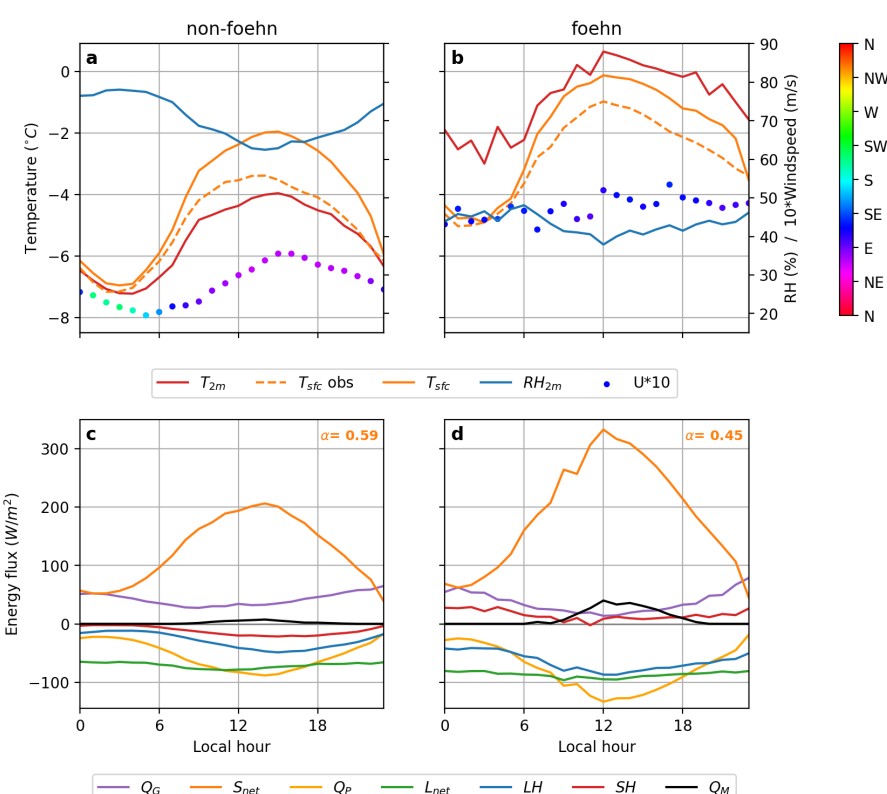

**Figure 10.** Average diurnal cycle in observed meteorological variables and simulated and simulated energy fluxes on days with and without foehn-conditions in the two melt periods of Figure 8 and 9. The average albedo during the non-foehn and foehn days are given in **c** and **d**
.





Foehn events also disturb the diurnal cycle in the summer sea-breeze circulation in the MDV. Under cloudless conditions, $R_{net}$ becomes negative at night and the cooled surface layer gives rise to down-glacier winds (Appendix C, fig C1). During daytime the down-glacier south-westerly flow is replaced by stronger north-easterly on-shore winds (Figure 10a). A similar diurnal wind regime in the Miers valley induced by night-time cooling through shading of the valley floor was observed with
eddy covariance and vertical wind profiles (Katurji et al., 2013). The diurnal wind regime is replaced by a steady south-easterly wind during foehn events (Figure 10b).

## 5   Discussion & Conclusions

We have presented an energy and mass balance time series from the ablation zone of a previously unstudied MDV glacier.
Glaciers in the MDV have been described as challenging for SEB modelling compared to higher melt environments. Daily variation in the energy fluxes is large, but daily mean values and melt residuals are typically small (Lewis et al., 1998). Therefore, errors in the measured radiation components can introduce large errors in the simulated melt energy. This study is the first SEB study in the MDV using data from a 4-component radiometer. Previous glacial energy balance modelling on MDV glaciers by Lewis et al. (1998) and Hoffman et al. (2008, 2014, 2016) have been limited to ablation measurements and thermis-
tor ice temperature measurements for model calibration and evaluation. The observations of outgoing longwave radiation at Joyce Glacier allowed for model evaluation against observed surface temperatures. The good agreement between simulated and observed surface temperatures with a RSME of 1.44 °C and a bias of 0.02 °C give us confidence in the calculated energy fluxes.

Solar radiation plays a dominant role in the energy balance of MDV glaciers. The measured incoming and net solar radiation
at Joyce are similar to values for Taylor Glacier (Table 2). While solar radiation penetration might play a small role in high-melt environments of temperate glaciers, Hoffman et al. (2008) showed the importance of including solar radiation penetration to accurately model the surface height change of MDV glaciers. Similarly, in this study we found that solar radiation penetration needed to be included in order to reproduce the observed surface temperatures at Joyce Glacier. It was estimated that 18% of $S_{in}$, equating to 43% of the $S_{net}$, penetrated the subsurface for the most optimal surface temperature simulation. This value is
comparable to 37% of $S_{net}$ that was found by Kuipers Munneke et al. (2009) for dry snow on top of the Greenland Ice Sheet. The percentage is smaller compared to the optimal simulations found by Hoffman et al. (2014) for Taylor Glacier using a re-freezing and drainage model, where 60% and 80% of $S_{net}$ penetrated the subsurface, respectively. In the simulation presented in this study, density does not change when subsurface melt runs off. Including the formation of such a 'weathering crust' in the surface height calculation as done by Hoffman et al. (2014) may enhance ablation and better represent observations, but
does not significantly affect the energy fluxes. Established methods to measure subsurface radiation, as well as internal melt rates and internal melt discharge for model calibration are still missing and subsurface penetration and melt remain, therefore, the largest uncertainties in MDV mass balance modelling.





**Table 2.** Comparison of geographic characteristics, meteorological conditions and SEB terms from glacial SEB studies in the MDV. The values represent averages for the December and January months within the study period and for Joyce averages over only hours with surface melt are given in square brackets.

|  | *Lewis et al (1998)* | *Hoffman et al 2008* | *Hoffman et al 2014 \** | *Hoffman et al 2014\** | *Hoffman et al 2014\** |  |
|---|---|---|---|---|---|---|
|  | **Canada** | **Taylor** | **Taylor** | **Taylor2** | **Canada** | **Joyce** |
| Years | '94 - '96** | '95 - '06 | '95 - '09 | '95- '09 | '95-'09 | '08 - '10 |
| Altitude (m asl) | 264 | 334 | 334 | 198 | 264 | 400 |
| Distance from coast (km) | 13 | 37 | 37 | 35 | 13 | 25 |
| $T_a$(°) | ∼ -3 | -3.0 | -3.0**** | -2.0**** | -3.3**** | -5.4 [-0.4] |
| RH (%) | n/a | 61.3 | n/a | n/a | n/a | 70.2 [51.8] |
| WS (m/s) | n/a | 4.4 | 4.3**** | 4.3**** | 3.0**** | 3.0 [3.2] |
| $\alpha$ (-) | 0.66 | 0.60 | n/a | n/a | n/a | 0.60 [0.48] |
| $S_{in}$ | 314 | 290 | n/a | n/a | n/a | 290 [607] |
| $S_{net}$ | 107 | 116 | ∼ 118 | ∼ 120 | ∼ 114 | 121 [315] |
| $Q_P$ | 0 | -20.9 | ∼ -94 | ∼ -93 | ∼ -93 | -53 [-135] |
| $Q_P/S_{net}$ | 0 | 0.18 | 0.80 | 0.79 | 0.76 | 0.44 [0.43] |
| $L_{net}$ | -68 | -65.7 | ∼ -62 | ∼ -57 | ∼ -66 | -69 [-98] |
| SH | -1 | 2.6 | ∼ 0 | ∼ +16 | ∼ -5 | -10 [-2.6] |
| LH | -21 | -37.3 | ∼ -38 | ∼ -65 | ∼ -26 | -29 [-48] |
| $Q_G$ | 0.5 | 6.3 | ∼80 | ∼ 82 | ∼ 71 | 42 [2] |
| $Q_M$ | 17*** | 1.0 | <1 | <1 | <1 | 2 [34] |

\* using drainage model

\*\* shows averages over period 21 Dec '94 - 21 Jan '95 and 22 Nov '95 - 10 Jan '96

\*\*\* 2.7 W/m$^2$ for '94/'95 and 46 W/m$^2$ for '95/'96 season

\*\*\*\* average over 2004-2009



The different ways to treat solar radiation penetration between the energy balance studies in the MDV are reflected in the
solar and subsurface heat fluxes (Table 2). When no penetration is allowed, the subsurface heat flux remains small, while more
heat is brought back to the surface for simulations with increasing rates of solar radiation penetration. The largest energy loss
for all MDV glaciers was by $L_{net}$ followed by $LH$. $LH$ losses to sublimation are smaller at Joyce compared to the more inland
Taylor Glacier, where humidity is lower and wind speeds generally higher (Table 2). The average $SH$ at most of the MDV
glaciers is slightly negative or positive (Table 2), while more negative at Joyce. The higher elevation and lower air temperatures
at Joyce Glacier ensure that energy losses through $SH$ are larger compared to the Taylor and Canada glacier, where the average
flux is small or positive closer to the glacier terminus. The diurnal pattern in $SH$ found in this study is comparable to a study
by Bintanja and van den Broeke (1994), that showed that unstable conditions (negative SH) during daytime lasted much longer
and nighttime conditions were less stable over blue ice compared to snow surfaces.

This study demonstrates the potential role of foehn events in enhancing melt amount and occurrence in the low-energy envi-
ronment of the MDV. Both the meteorological observations and AMPS output are used for detection of foehn events at Joyce
Glacier. Foehn occurred at Joyce Glacier for 15% of the days, which is on average lower compared to foehn occurrence in
Taylor and Wright valleys, although this frequency has a large inter-annual variability (Speirs et al., 2013). As foehn condi-
tions at AWS sites in all MDV glaciers are related to cyclone activity over the Ross Sea Region, differences in occurrence arise
from the wind patterns within the complex terrain of the MDV. The location of the low pressure system and the direction of
the associated flow determines where foehn conditions occur in the MDV. When the southerly flow is forced through the gap
north of the Royal Society Range, foehn is observed in the northern MDV, with the mechanism described by Steinhoff et al.
(2013, 2014). When the flow is blocked at low-levels by the Royal Society Range and air is sourced from higher potentially
warmer air, this leads to foehn conditions in the south-eastern MDV through isentropic drawdown. This study shows the com-
plex interaction of the synoptic flow with the topography of the MDV. Foehn occurrence at Joyce Glacier during this 14-month
study might not be representative for the average foehn conditions and therefore future studies on the topographic-forced flow
dynamics in the complex MDV terrain are needed to understand the relative occurrence of foehn events in the different MDV.

The relative humidity dropped on average by 40% during foehn events at Joyce Glacier, with the lowest humidity reaching
23%. Even lower relative humidity values close to 12% were observed by Speirs et al. (2010) during foehn events in 2007 at
Commonwealth, Canada and Howard glaciers and between 7% and 17% at non-glacial MDV sites. Annual average relative hu-
midity at Joyce Glacier and previously mentioned MDV glaciers are comparable (Obryk et al., 2020), suggesting foehn drying
might be weaker at Joyce Glacier. As shown for the MDV, foehn can bring air temperatures above zero, leading to temperatures
of up to 5.8°C at Joyce Glacier. The largest increase in air temperature is 17.9°C during foehn at Joyce Glacier, compared to
increases of 25°C found for a 2007 foehn event at Howard, Commonwealth and Canada glaciers (Speirs et al., 2010). AWS
sites in valley floors even showed temperature increases of up to 50°C in the MDV during foehn events (Speirs et al., 2010). In
the Taylor, Wright and Victoria Valleys, winters are characterized by cold air pool formation under stable conditions. When this
cold air is eroded by foehn air it leads to large temperature changes. Cold pool formation is likely weaker at Joyce Glacier due





to its higher location in a valley, where down-glacier winds are more frequently and prevent such stable conditions. Whether

the magnitude in foehn warming at MDV glaciers are associated with the difference in the underlying foehn mechanism and

its ability to penetrate down to the ice surface should be further explored in future studies.

During the two melt seasons in the AWS record at Joyce, melt occurred more frequently during foehn events compared

to non-foehn conditions and melt energy was on average larger during foehn melt events. This agrees with findings of foehn-

induced melt over the Antarctic Peninsula. There, the highest melt rates occur during foehn conditions and foehn can contribute

to 45% of the total melt in some regions of the Larsen C shelf (Elvidge et al., 2020). The extra shortwave radiation through

widespread cloud-clearance during foehn events over the Antarctic Peninsula (King et al., 2017; Elvidge et al., 2020; Turton

et al., 2020; Laffin et al., 2021) was not evident from the summer foehn events at Joyce Glacier. In fact, during winter foehn

events, Joyce Glacier experienced on average cloudier conditions. The foehn warming over Joyce is dominated by isentropic-

drawdown and foehn-clearance might play a minor role through the strong low-level blocking by the Royal Society Range,

which prohibits the warmer and more humid air from low-levels from rising and condensing on the windward side (Zou et al.,

2021b). Besides, the foehn winds might not be able to penetrate fully to the surface over Joyce glacier and might only partly

erode the moist air originating from McMurdo Sound. As melt in the MDV is driven by solar radiation, future foehn studies in

the MDV should investigate the cloud effects during summer foehn in the MDV. There is a need for better radiation observa-

tions at the MDV AWS sites to study the sensitivity of melt with respect to clouds.

Over the Antarctic Peninsula, especially under strong foehn jets, melt can be $SH$ driven leading to melt even in the polar

night (Kuipers Munneke et al., 2018; Elvidge et al., 2020). While this study confirms the increased $SH$ warming and $LH$ cool-

ing during foehn events, melt remains always solar radiation dominated at Joyce Glacier. The strong increase in $SH$ warming

is largely compensated by increased $LH$ losses at Joyce Glacier, and the increased melt rates under foehn conditions more

likely result from lowered surface albedo and therefore increased $S_{net}$. Distinctive for the MDV glaciers is the change from

surface melt with sub-zero air temperatures during non-foehn conditions to melt under stable surface layer conditions during

foehn events. As melt at Joyce Glacier is driven by $S_{net}$, it occurs solely during midday when solar fluxes are peaking. The

increased $SH$ heating throughout the entire day during foehn events suggests that melt does also occur in periods with a lower

solar angle, under the condition that $SH$ is not compensated by large $LH$ losses. Studies covering several melt seasons are

needed to confirm the impact of foehn events on melt duration and intensity in the MDV.

While melt during foehn events is still a small part of the total melt, we suspect that foehn events play an important role in

melt generation in the MDV as the largest melt rates are found during or just after foehn events. At this point the ice surface has

warmed and clear-sky conditions lead more frequently to melt generation compared to pre-foehn days. We hypothesize that

foehn in this way 'primes' the surface through warming and albedo reduction for melt generation in the MDV. Since summer

temperatures are mostly below zero in the MDV, foehn occurrence likely plays a key role to reach the melting point through

this surface 'priming'. This highlights the different role of foehn events in melt generation in the colder environment of the



MDV compared to the Antarctic Peninsula. Also, above freezing temperatures on the Ross Ice Shelf were essential for melt
under foehn-conditions (Zou et al., 2021b).

    This study is the first to study the foehn-melt interaction in the MDV from an energy balance perspective, and more studies
including spatial and temporal variability are needed to confirm the role of foehn occurence in melt generation. Foehn induced
melt at the MDV might spatially vary through the interaction between the synoptic flow with the local topography and through
localized melt at topographic features such as ice cliffs (Lewis et al., 1999). A climatological study of foehn occurrence and
foehn strength at the MDV glaciers is needed to better understand the impact of temporal variability of foehn strength and
duration on melt occurrence and melt rates. Dubnick et al. (2017) showed strong nutrient-specific responses to changes in
runoff of Joyce Glacier and therefore future studies using a 'watershed' approach would provide key data on how changes
in the frequency or duration of foehn events may be disproportionately impacting the wider receiving environment through
enhanced hydrological and associated biogeochemical change.

*Author contributions.* MGH performed most of the data processing and analyses. JFO and NJC collected the observational data. MGH, NJC,
CH, MB and MW were involved in the conceptualization and methodology. MGH wrote the draft-paper and edits with review and editing
by all co-authors.

*Competing interests.* The authors declare that they have no conflict of interest.

*Acknowledgements.* This data collection was supported by Antarctica New Zealand logistics funding for event K064 to S. Fitzsimons and
J. F. Orwin. This research was partially supported by the New Zealand Antarctic Science Platform (ANTA1801, program: Projecting Ross
Sea Region Ecosystem Changes in a Warming World), and the Royal Society of New Zealand (Grant No. RDF-UOC1701). We thank
Mark Seefelt for making the AMPS data available from their data archive. The AMPS data was processed and stored on the New Zealand
eScience Infrastructure (NeSI) high performance computing facilities. We would like to thank the WRF-python community (Ladwig, 2017)
and Michaela Sizemore for wrf-python support for analysis of the atmospheric model data presented in this study.



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





**Appendix A: AWS sensors**

**Table A1.** Specifications of sensors installed on the Joyce AWS.

| Variable | Instrument | Range | Accuracy |
|---|---|---|---|
| Air temperature | Vaisala HMP45A | −40 to +60°C | ± 0.2 °C at 20 °C |
| Relative humidity | Vaisala HMP45A | 0–100% | 2% (RH 0–90%) at 20 °C |
| | | | 3% (RH 90–100%) at 20 °C |
| Wind speed | R.M. Young 05103 | 0–100 ms-1 | ± 0.3 ms-1 or 1% |
| Wind direction | R.M. Young 05103 | 0–360° | ± 5° |
| Air pressure | Setra CS100 270 | 600–1100 mb | ± 0.5 mb at 20 °C |
| Shortwave radiation | Kipp and Zonen CNR1—CM3 | 0.31–2.8 $\mu$m | EADT ± 10% |
| Longwave radiation | Kipp and Zonen CNR1—CG3 | 5–50 $\mu$m | EADT ± 10% |
| Surface height | CSI SR50A | 0.5–10 m | 0.01 m or |
| | | | 0.4% of distance to target |

**Appendix B: Foehn detection**

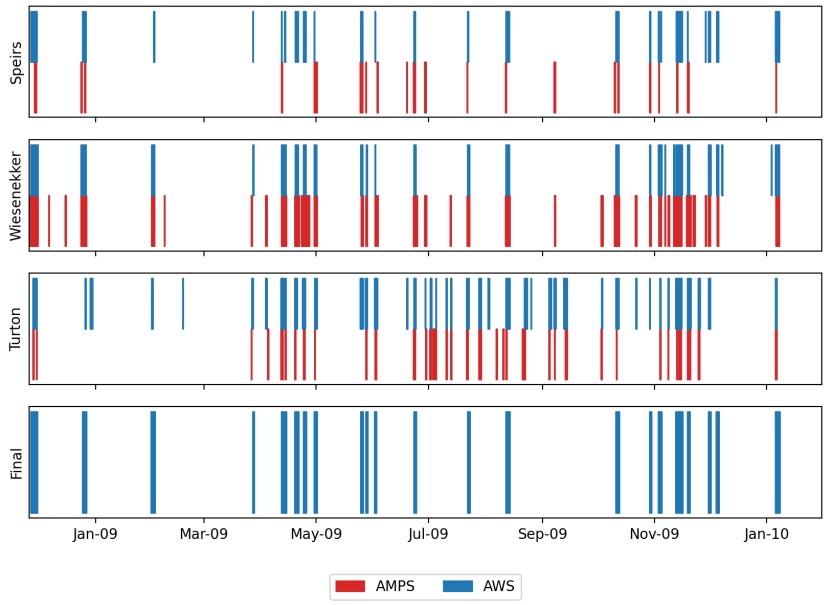

**Figure B1.** Foehn detected by the Speirs, Wiesenekker and Turton method in AWS (blue) and AMPS data (red) at Joyce Glacier. The final detection shows where 2 out of 3 methods agree in a 6 h period based on AWS data.





## Appendix C: Diurnal cycle on cloud and overcast days

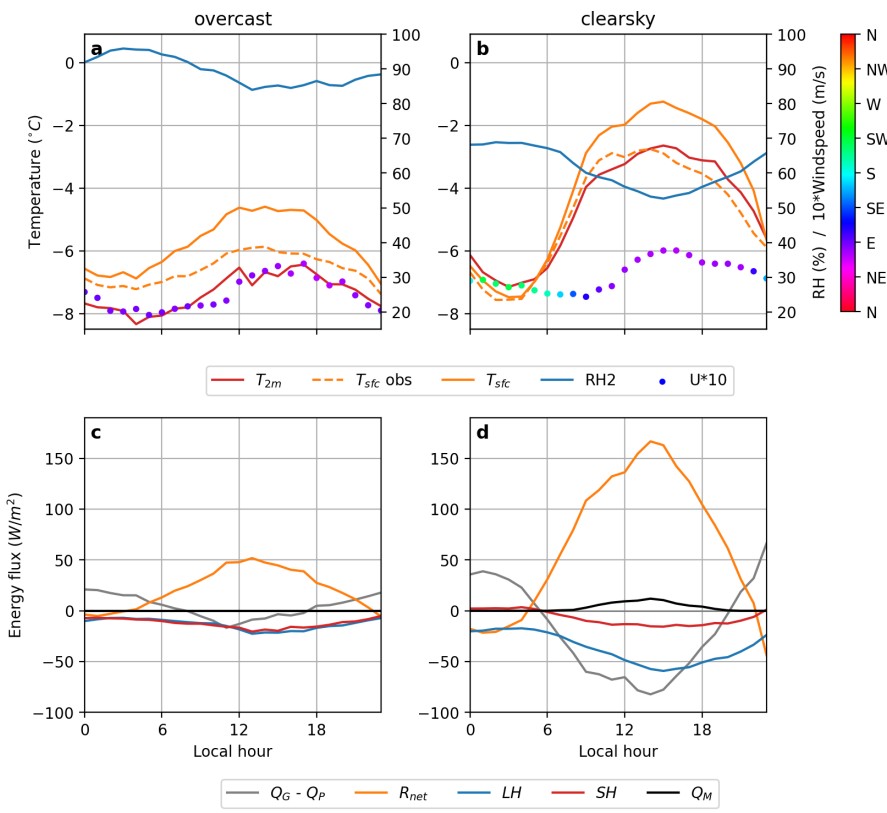

**Figure C1.** Average diurnal cycle in observed meteorological variables and simulated and simulated energy fluxes on overcast ($N_\epsilon > 0.7$) and clear-sky ($N_\epsilon < 0.3$) days in December an January. $R_{net}$ represents $S_{net} + L_{net}$.