# Peer review of "The surface energy balance during foehn events at Joyce Glacier, McMurdo Dry Valleys, Antarctica"

_The Cryosphere, 2022_

## Referee Comment (RC1)

Review of Hofsteenge et al. 2022 The surface energy balance during foehn events at Joyce Glacier, McMurdo Dry Valleys, Antarctica

The impact of foehn events on glaciers/melting has gained traction in the last decade, with many studies documenting the link between foehn events and melting in Antarctica, using a variety of methods. That being said, this paper contains novel aspects: this glacier is largely unstudied, especially in the SEB and foehn field, and reveals a more complicated story of the mechanisms behind foehn in this region, thereby progressing our knowledge. The study investigates a 14-month period, using both observations from an automatic weather station and output from a high resolution model. Perhaps some of the highest spatial resolution output used for foehn studies (in this region) thus far. The authors employ a combination of previously developed methods, highlighting an overall good agreement between methods, and between observations and model. A SEB model is used to look at the impact of foehn on various components of the energy balance as well as melting. The authors are aware of the limitations of their study and don't oversell their results or conclusions. They could actually highlight further some aspects of novelty which are only mentioned in the discussion. There are some areas of clarification required for the method, but I don't expect this to change any results or conclusions. The paper is structured well and is nice to read, with some useful and easy to interpret figures. Overall, I suggest **minor** corrections and outline these below.

**Page 2, Lines 29-38:** Is it possible to include more details on the map of the Antarctic in Figure 1? You mention a number of mountain ranges and glaciers throughout the paper, but people unfamiliar with the area will find it difficult to put these into place. This is especially important when you discuss the interaction of the low pressure systems and airflow for the two different foehn mechanisms. I would suggest either 4 panels, with one providing more details (e.g Transantarctic mountain range, Taylor glacier), or exchanging the AWS plot with an additional map.

**Page 2, Line 34:** Include a brief description of what a foehn wind is. TC is quite an interdisciplinary journal, but still read mostly by glaciologists who might be unfamiliar with the more atmospheric terms.

**Page 3, Line 57:** Similar to above but for sensible heat, or perhaps just include *(through warmer and gustier than average winds)* after the words 'sensible heating'.

**Introduction general:** I think there are some aspects of novelty that you could highlight more in the introduction. Authors should double check that the following statements are true before including them, but if so, you could include them in your introduction. Is the 1.67km horizontal resolution of AMPS the highest spatial resolution that has been used for foehn studies in the Antarctic so far? Elvidge et al (2015) and Turton et al. (2017) have stressed the importance of higher resolution modelling for foehn winds due to their complex interaction with the topography. But most AMPS studies I can think of use 5km or coarser resolution for foehn studies. The higher resolution doesn't necessarily imply better results though, as the wind direction problem could be associated with this. You mention in the discussion that this is the first time a 4-component radiometer is used in the MDVs.

Which was important for your surface temperature calculations, which are also not often included in foehn studies.

**Page 4, Line 90-95:** When you list your observations and calculations, you don't mention surface temperature.

**Page 6, Line 137:** Include a reference for the 12- hour spin up and removal. There can be quite a debate about the length of time that should be used for spin up, so a reference stating that this comes from the operational use of AMPS (and therefore isn't your decision) would strengthen the sentence.

**Page 6, Line 150:** What resolution was the AMPS that Speirs used? On line 159 you explain the wind direction difficulties and therefore use a different direction threshold for AWS and AMPS. Was this also done by Speirs, or is this your decision? Does the choice of a different threshold affect when foehn is detected, and how did you settle on the specific directions you use?

**Page 7, Line 184:** Include in this sentence that these criteria are applied to AWS near-surface data.

**Page 8, Line 190:** Include that these criteria are applied to AMPS data.

**Method general:** You need to provide more clarity on your foehn criteria that you employ after the evaluation stage and on what criteria you apply to near-surface conditions from AMPS and which to upper-air data (isentropic detection). For example, did you apply the methods to detect near-surface changes (Speirs, Wiesenekker and Turton Part 1) to AMPS near-surface data and AWS data? If so, is this appropriate, given that the Turton Part 2 method should be used for AMPS data and not Part 1 method? AMPS was shown to relatively poorly represent near-surface conditions in foehn, and therefore a specific algorithm for the isentropic drawdown was used in AMPS. Or did you apply the Speirs and Wiesenekker method to AMPS near-surface and the Turton Part 2 method to AMPS upper air conditions? This becomes important in Page 10, line 231 where you compare the number of foehn in AWS and AMPS. Turton method shows the biggest difference in AMPS and AWS and you put that down to weaker humidity drops in AMPS, however, the original Turton method didn't look at the near-surface conditions in AMPS, but rather the isentropic drawdown to define a foehn. It's fine for you to adapt the methods for your own use/different location, but I would make it clear that what exactly you extracted from AMPS and how you applied the criteria.

Did you look at wind speed bias between AWS and AMPS to assess whether the 5 m/s Speirs and 4 m/s Wiesenekker is appropriate in AMPS if there was a considerable bias?

You say that at least 2 out of 3 methods must detect foehn for you to use it – but is this in AWS only data, or AWS and AMPS data? If just AWS, it needs to be a little clearer that the AMPS results are only used for evaluation.

**Foehn detection and model evaluation**

Some comments from this section are include in the above paragraph.

**Page 11, section 3.2:** Is there a reason you decided not to use the AMPS SEB output for an additional analysis, such as in King et al. (2015, 2018, and others)? I was occasionally confused by the word 'simulated' for the SEB section, as it comes after the use of AMPS for many paragraphs. The SEB model is run using observations, so it is more observationally forced than simulated. I wonder if the word 'simulated' should be linked only to AMPS to reduce this confusion. I'll leave this up to the authors however, as perhaps my previous use of SEB output from AMPS has skewed the way I am reading this.

How did you create the ensemble? How many runs is this and what initial variable or value did you alter?

**Page 12, Line 285:** Is the anomaly plot from AMPS? Include in figure caption or text to make it clear.

**Page 13, Line 291:** Some important places for context here but no map to point them out.

**Page 14, Line 305:** If you applied the Turton Part 2 algorithm to AMPS data, to detect isentropic drawdown, this could be the reason for a higher number of foehn events using this method than the other two. If Joyce Glacier is more susceptible to isentropic drawdown, the Turton method is more likely to pick that up than the methods using only near-surface characteristics.

**Page 14, line 315:** What is the *(increasing)* for here?

**Page 14, line 325:** This sentence is a little confusing, you write both simulated and observed surface temperatures twice, perhaps shorten the sentence or remove the 'based on both simulated and observed surface temperatures'?

**Page 15, line 339:** Do you detect/categorise katabatic winds too, or you just look at the main characteristics to decide that during non-foehn conditions, the SH flux is due to katabatics? Can we be sure that we are not falsely categorising katabatics as foehn winds? Or vice versa? Especially with the Wiesenekker method, which doesn't take into account temperature or relative humidity.

**Page 16, line 352:** Is this statement about no MDV melt during cloud cover your result (so specific to Joyce Glacier during the 14 month period) or is this information from elsewhere and more of a generalisation. As figure C implies your data alone, but by saying 'MDV glaciers' it seems like you have more evidence for this statement.

Nice figures and use of space on the figures.

**Page 17, line 365:** There has been a study that looks at the longer-term effect of foehn warming on the snowpack, or pre-conditioning of the surface for future melt. E.g Kuipers Munneke et al. 2014, Elvidge et al. 2020. Could include reference to similar studies also on page 23, around line 505.

**Page 17, line 377:** include *'during such conditions'* after *'…key role in melt occurrence'*.

**Page 18, line 379:** Did you look into the melt immediately after foehn events? 6-12 hours after the foehn event for example? It could provide a nice extra result.

**Page 18, line 382:** In the brackets for the figure citation, include that the reader should look for the grey line on 9a, as it took me a while to see it amongst the other colours and bars.

**Page 19, line 397:** include *'in which'* after *'68% of the hours'*

**Page 19, line 398:** change *'is peaking'* to *'peaks'*

**Page 19, figure 10:** Consider changing the colourbar to one that isn't rainbow, as this can be difficult for those with colourblindness. Interesting way to represent wind direction – I like it! Same for figure C1.

**Page 20, line 410:** is this steady south-easterly a product of your foehn algorithm though?

**Page 20: first paragraph of section 5:** Some of this novelty should be clearer in the introduction.

**Page 21, line 453:** You also had quite a short duration of data, so couldn't look at the interannual variability or be sure that this 15% value is characteristic for your region either.

**Page 21, line 460:** The relatively coarse vertical resolution of AMPS (44 levels) may also not allow simulation of conditions closer to the surface. Many regional models now have 60-70 vertical levels.

---

## Referee Comment (RC2)

**Review of Hofsteenge et al. (2022): The surface energy balance during foehn events at Joyce Glacier, McMurdo Dry Valleys, Antarctica**

**Overall**
This study investigates the foehn and surface energy balance (SEB) over the Joyce Glacier, where receives less attention in the previous research. This paper use SEB model simulations that driven by the AWS observation and AMPS forecast outputs. Authors suggest that SWnet is the dominant driver during the daytime with a continuous contribution from sensible heat flux. Surface sublimation offsets the positive SEB, and reduced albedo further enhances the melting. This paper also highlights the important of solar radiation penetration for the accurate estimation of SEB. This paper includes several novel findings, and the scientific part is solid. A few results will be more convincing with more data or better explanations. The Discussion section might need some reorganizations. Thus, I would suggest a **minor revision** for this paper.

**General**
   A. Structure:
   The conclusion and discussion part includes lots of interesting and novel findings. I think the reader will appreciate that if it is better organized. Authors first talked about SEB, then back to foehn (which contributes to the increase SWD via foehn clearance, SH via turbulence/downslope wind on the leeside, etc). The comparison between Joyce Glacier and AP is great. However, the connection between each topic is kind of loose.

   B. Data:
   AMPS provides reliable forecast data for operational use. However, it has difficulties simulating the cloud conditions partially due to the microphysical scheme it is using (WSM5). This can lead to cold bias on the surface and compromise its ability to describe the SEB. Has author evaluated AMPS data with reanalysis data like ERA5 (31km) or ERA5 Land (9km)? It is not a serious problem, since this paper mainly analyzes SEB based on the SEB model. The surface variable (2mT, 10m Wind) needs a brief evaluation observation or reanalysis data. Also, can author explain why using a 12-h spin up instead of 24-h (more common)?

   The cloud condition is important to back up some conclusions in this paper. I would suggest author include cloud observation from satellite or simulation from reanalysis data (e.g., MODIS, ERA5, CERES, etc.) to better explain the SEB conditions (e.g., Ln 366).

**Minor:**
   1. Echoing another reviewer's suggestion. Need more labels for the local topographic features in Fig. 1. Plus, the current labels are also hard to see.
   2. Terms like foehn warming, isentropic drawdown, internal melting might need some brief description.
   3. I would suggestion authors use forecast output/forecast for AMPS, and simulation for SEB model outputs.
   4. Are you able to differentiate the foehn wind with katabatic wind in your study?

Ln 300: Have authors checked the vertical wind profile or calculated the Fr number to confirm the low-level blocking? How strong the blocking is?

Ln 385: Some satellite or reanalysis data can be used here to provide some cloud information, such as cloud cover, phase, height, and thickness. Also, I am curious if there is any cloud formation, where is the moisture source?

Ln 425: Does the solar radiation penetration associated with the duration of the clear-sky conditions? Will the accurate description of the changing surface (wet snow, melting pond, refreeze) benefit the estimation of solar radiation penetration?

Ln 475: I think whether the downslope wind is able to reach the ice surface is highly associated with the mountain waves on the leeside, stationary orographic gravity, hydraulic jump, or breaking waves? This is not only associated with the magnitude of foehn, but also the detailed mechanism.

Fig. 5 the wind vector reference is hard to see. Which AMPS domain is used here (02?) at what resolution (15km)? what is the baseline for the anomaly? I would suggest including those info in the caption.

Fig 10&C1. Just a suggestion. Will that be better to use wind barbs to indicate the direction instead of different colors?

---

## Author Comment (AC1)

**Comments on reviewer #1**, the original review is given in black, and author comments in **blue.**

**Review of Hofsteenge et al. 2022 The surface energy balance during foehn events at Joyce Glacier, McMurdo Dry Valleys, Antarctica**

The impact of foehn events on glaciers/melting has gained traction in the last decade, with many studies documenting the link between foehn events and melting in Antarctica, using a variety of methods. That being said, this paper contains novel aspects: this glacier is largely unstudied, especially in the SEB and foehn field, and reveals a more complicated story of the mechanisms behind foehn in this region, thereby progressing our knowledge. The study investigates a 14-month period, using both observations from an automatic weather station and output from a high resolution model. Perhaps some of the highest spatial resolution output used for foehn studies (in this region) thus far. The authors employ a combination of previously developed methods, highlighting an overall good agreement between methods, and between observations and model. A SEB model is used to look at the impact of foehn on various components of the energy balance as well as melting. The authors are aware of the limitations of their study and don't oversell their results or conclusions. They could actually highlight further some aspects of novelty which are only mentioned in the discussion. There are some areas of clarification required for the method, but I don't expect this to change any results or conclusions. The paper is structured well and is nice to read, with some useful and easy to interpret figures. Overall, I suggest **minor** corrections and outline these below.

We would like to thank you for taking the time to review this manuscript. We are happy to read your positive comments and to incorporate the suggestions you made to improve the paper. We describe below in more detail which changes have been made to the manuscript.

**Page 2, Lines 29-38:** Is it possible to include more details on the map of the Antarctic in Figure 1? You mention a number of mountain ranges and glaciers throughout the paper, but people unfamiliar with the area will find it difficult to put these into place. This is especially important when you discuss the interaction of the low pressure systems and airflow for the two different foehn mechanisms. I would suggest either 4 panels, with one providing more details (e.g. Transantarctic mountain range, Taylor glacier), or exchanging the AWS plot with an additional map.

We agree that Figure 1 could be more informative with the locations that are mentioned in the paper, therefore we made a new figure that includes all the locations and topographic features that are discussed in the foehn description.

**Page 2, Line 34:** Include a brief description of what a foehn wind is. TC is quite an interdisciplinary journal, but still read mostly by glaciologists who might be unfamiliar with the more atmospheric terms.

We added a sentence describing foehn winds: 'Foehn winds typically occur through topographic modification of flow in the lee of mountain barriers, resulting in strong and warm winds.'

**Page 3, Line 57:** Similar to above but for sensible heat, or perhaps just include *(through warmer and gustier than average winds)* after the words 'sensible heating'.

We have included here: 'through an increase in turbulent mixing resulting from gustier and warmer wind conditions.'

**Introduction general:** I think there are some aspects of novelty that you could highlight more in the introduction. Authors should double check that the following statements are true before including them, but if so, you could include them in your introduction. Is the 1.67km horizontal resolution of AMPS the highest spatial resolution that has been used for foehn studies in the Antarctic so far? Elvidge et al (2015) and Turton et al. (2017) have stressed the importance of higher resolution modelling for foehn winds due to their complex interaction with the topography. But most AMPS

studies I can think of use 5km or coarser resolution for foehn studies. The higher resolution doesn't necessarily imply better results though, as the wind direction problem could be associated with this. You mention in the discussion that this is the first time a 4-component radiometer is used in the MDVs. Which was important for your surface temperature calculations, which are also not often included in foehn studies.

Thanks for your suggestions. Regarding the AMPS resolution, the mechanism resulting in foehn effects in the northern MDV has been studied with Polar WRF at a higher resolution than AMPS (0.5km nested grid scales). We have added in the introduction the following statement to clarify what model resolution has been used before: 'Steinhoff et al. (2013) used the polar version of the Weather Research and Forcasting (WRF) model on a 0.5km nested domain to study physical processes responsible for foehn events in Taylor, Wright and Victoria Valley.'

We did include as you suggested in the Introduction that a 4-component radiometer was used for the first time for a SEB study in the Dry Valleys, which improved the simulation of the surface temperatures. The revised text is: 'This is the first SEB study in the MDV to use a glacier based 4-component radiometer, which allowed for calibration of the simulated surface temperatures and increases accuracy of the simulated energy fluxes.'

**Page 4, Line 90-95:** When you list your observations and calculations, you don't mention surface temperature.

To clarify this we have moved the sentence that Tsfc is calculated from outgoing longwave radiation up to this section describing the observations.

**Page 6, Line 137:** Include a reference for the 12- hour spin up and removal. There can be quite a debate about the length of time that should be used for spin up, so a reference stating that this comes from the operational use of AMPS (and therefore isn't your decision) would strengthen the sentence.

Since the full AMPS output is not available anymore in an online archive, we have used a thinned AMPS product that was saved as 12-24h forecast hours. We have now cited a paper using AMPS in the same way (Seefeldt and Cassano, 2012), who provided us these AMPS products.

**Page 6, Line 150:** What resolution was the AMPS that Speirs used? On line 159 you explain the wind direction difficulties and therefore use a different direction threshold for AWS and AMPS. Was this also done by Speirs, or is this your decision? Does the choice of a different threshold affect when foehn is detected, and how did you settle on the specific directions you use?

In the study by Steinhoff et al. (2014) the same wind-direction criteria were used on AWS and AMPS output from a 2km nested domain. AMPS used in this study has a higher resolution and we think that the difference in wind direction between AMPS and AWS observations is caused by the more complex valley system around Joyce glacier compared to the wider valleys studied in Steinhoff et al. (2014). We have now clarified why a different wind direction threshold was used in the explanation of the Speirs method as follows: 'A different wind direction criterion is used on AWS data compared to AMPS forecast, since the weather station was located close to the valley side and AMPS does not capture well the topographic modification of the winds (Sect. 3.1)'. The wind-direction thresholds were chosen to represent the expected foehn wind direction and to make sure that it does not include the sea-breeze wind direction.

**Page 7, Line 184:** Include in this sentence that these criteria are applied to AWS near-surface data.

We have applied the near-surface conditions of Turton Part 1 as well using AMPS forecast for testing purpose. Under your comment below on 'Method general' we have included how we have made this clear in the manuscript.

**Page 8, Line 190:** Include that these criteria are applied to AMPS data.

Thank you, we have clarified this as follows: 'Sufficient isentropic drawdown, determined as follows from AMPS forecast (Fig. 2):'.

**Method general:** You need to provide more clarity on your foehn criteria that you employ after the evaluation stage and on what criteria you apply to near-surface conditions from AMPS and which to upper-air data (isentropic detection). For example, did you apply the methods to detect near-surface changes (Speirs, Wiesenekker and Turton Part 1) to AMPS nea2r-surface data and AWS data? If so, is this appropriate, given that the Turton Part 2 method should be used for AMPS data and not Part 1 method? AMPS was shown to relatively poorly represent near-surface conditions in foehn, and therefore a specific algorithm for the isentropic drawdown was used in AMPS. Or did you apply the Speirs and Wiesenekker method to AMPS near-surface and the Turton Part 2 method to AMPS upper air conditions? This becomes important in Page 10, line 231 where you compare the number of foehn in AWS and AMPS. Turton method shows the biggest difference in AMPS and AWS and you put that down to weaker humidity drops in AMPS, however, the original Turton method didn't look at the near-surface conditions in AMPS, but rather the isentropic drawdown to define a foehn. It's fine for you to adapt the methods for your own use/different location, but I would make it clear that what exactly you extracted from AMPS and how you applied the criteria.
Did you look at wind speed bias between AWS and AMPS to assess whether the 5 m/s Speirs and 4 m/s Wiesenekker is appropriate in AMPS if there was a considerable bias?
You say that at least 2 out of 3 methods must detect foehn for you to use it – but is this in AWS only data, or AWS and AMPS data? If just AWS, it needs to be a little clearer that the AMPS results are only used for evaluation.

Thank you for your extensive feedback here, that is very helpful. We have made several adjustments to clarify when AWS and AMPS data are used and for which purpose. We have used as in Turton 2018 Part I on near-surface conditions in combination by Part II on AMPS upper air forecast. We tested all methods using both near-surface conditions based on AWS observations and AMPS forecast. We have clarified that the Turton method was designed to combine AWS observations and AMPS forecast for the atmospheric structure in the description of the method: 'Turton et al. (2018) developed a more complex classification scheme where a combination of criteria to detect observed near-surface foehn conditions and AMPS forecasted foehn characteristics in the upper-air are used for foehn detection over the Antarctic Peninsula.'

We clarified that we test the method here as well with AMPS near-surface conditions in Section 2.4.3 as 'For evaluation purposes, this method is also tested using AMPS forecasted near-surface conditions in Sect. 3.1' . In Section 3.1 we have clarified again that AMPS near-surface conditions are only used for testing and in the remainder of the manuscript AWS near-surface conditions are used with the following two sentences: 'We applied the 3 methods described in Sect. 2.4 to identify foehn conditions at the Joyce Glacier using both near-surface conditions based on AWS observations and AMPS output for the 14-month study period.' and 'Throughout the rest of this paper a combination of the detection methods using near-surface conditions from AWS observations are used to classify foehn.'

Regarding AMPS bias of wind direction, we found on average a slight negative bias in AMPS, which is due to a negative bias under calm conditions (poor representation of down-glacier winds, Sect 3.1). During strong wind events, AMPS often has a positive bias and therefore we think the AMPS windspeed bias does not impact the foehn detection.

**Foehn detection and model evaluation**
Some comments from this section are include in the above paragraph.

**Page 11, section 3.2:** Is there a reason you decided not to use the AMPS SEB output for an additional analysis, such as in King et al. (2015, 2018, and others)? I was occasionally confused by the word 'simulated' for the SEB section, as it comes after the use of AMPS for many paragraphs. The SEB model is run using observations, so it is more observationally forced than simulated. I wonder if the word 'simulated' should be linked only to AMPS to reduce this confusion. I'll leave this up to the authors however, as perhaps my previous use of SEB output from AMPS has skewed the way I am reading this.

To solve the confusion around AMPS and SEB simulations, we followed the suggestion by reviewer #2 to change all references to the AMPS data by using 'forecast' and using 'simulations' for the SEB results. We have considered to use the AMPS SEB output to discuss spatial SEB patterns, however we found that it poorly represented the SEB of the dry valley glaciers. The limitations of AMPS are mainly that i) the snow model does not allow solar penetration, ii) AMPS solar radiation poorly captures the variability through cloud cover and iii) the sensible heat flux did not capture the typical unstable conditions.

How did you create the ensemble? How many runs is this and what initial variable or value did you alter?

The ensemble was created using the combinations of parameter values listed in Table 1. To clarify we have added a reference to this table when discussing Figure 4 showing the ensemble spread.

**Page 12, Line 285:** Is the anomaly plot from AMPS? Include in figure caption or text to make it clear.

This has now been included in the captions of figures showing AMPS forecast output.

**Page 13, Line 291:** Some important places for context here but no map to point them out.

We have added a reference to Figure 1, which now is adjusted to show all the important places (see answer to previous comment).

**Page 14, Line 305:** If you applied the Turton Part 2 algorithm to AMPS data, to detect isentropic drawdown, this could be the reason for a higher number of foehn events using this method than the other two. If Joyce Glacier is more susceptible to isentropic drawdown, the Turton method is more likely to pick that up than the methods using only near-surface characteristics.

Thank you for this suggestion, this could indeed be the case. It might be that Joyce is occasionally influenced by drying & isentropic warming, while windspeeds do not remain elevated and are therefore not picked up by the other methods.

**Page 14, line 315:** What is the *(increasing)* for here?

Changed into 'March and October can be considered as transition months with a decreasing or increasing contribution of solar radiation', to clarify in March solar radiation decreases and in October it increases.

**Page 14, line 325:** This sentence is a little confusing, you write both simulated and observed surface temperatures twice, perhaps shorten the sentence or remove the 'based on both simulated and observed surface temperatures'?

We have rephrased this sentence to simplify and clarify it, as follows: 'Although simulated surface temperatures have a positive bias in summer (Sect. 3.2), both simulated and observed surface temperatures are often warmer than the summer air temperature, confirming the often negative sign of SH'.

**Page 15, line 339:** Do you detect/categorise katabatic winds too, or you just look at the main characteristics to decide that during non-foehn conditions, the SH flux is due to katabatics? Can we be sure that we are not falsely categorising katabatics as foehn winds? Or vice versa? Especially with the Wiesenekker method, which doesn't take into account temperature or relative humidity.

We believe that we do not falsely detect katabatic or other non-foehn wind events, since we use a combined method where two out of the three methods need to flag the 6h period as foehn. In practice this means that either the Turton method or the Speirs method confirms the isentropic drawdown or the surface warming & drying characteristic of the foehn event. We realize that adiabatic warming of katabatic winds could result in a similar temperature and relative humidity signature. However, the wind events are unlikely to be of a katabatic nature as they are associated with cyclone activity and synoptic flow that leads to foehn events rather than air originating from the Antarctic plateau. Please note that we have changed the structure of the Discussion & Conclusion section (as suggested by reviewer 2) and have included more discussion around the katabatic vs foehn origin of these winds here to clarify this point.

**Page 16, line 352:** Is this statement about no MDV melt during cloud cover your result (so specific to Joyce Glacier during the 14 month period) or is this information from elsewhere and more of a generalisation. As figure C implies your data alone, but by saying 'MDV glaciers' it seems like you have more evidence for this statement.

We rephrased this statement, to clarify that previous research has shown that MDV glacial melt is solar radiation driven and that in this study we find that melt does very rarely occur during cloudy conditions, as follows: "Melt at MDV glaciers is driven by solar radiation (Hoffman et al., 2008) and melt does rarely occur during days with clouds in the 14-month record at Joyce (Appendix C, Fig. C1). 83% of the surface melt at Joyce Glacier occurred during clear-sky conditions (Ne < 0.2), and only 0.5% during overcast conditions Ne > 0.8).".

Nice figures and use of space on the figures.

**Page 17, line 365:** There has been a study that looks at the longer-term effect of foehn warming on the snowpack, or pre-conditioning of the surface for future melt. E.g Kuipers Munneke et al. 2014, Elvidge et al. 2020. Could include reference to similar studies also on page 23, around line 505.

Thanks for these suggestions. We have included the following references on the pre-conditioning effects found by these studies in the Introduction and Discussion: 'Here, foehn events cause the highest melt rates and the increased melt associated with foehn plays a preconditioning role in ice shelf melt and instability (Kuipers Munneke et al., 2014)' and 'However, in a similar way foehn plays a major role in prolonging the melt season at Larsen C ice shelf and spring foehn events precondition the ice shelf for enhanced summertime melt via reduced albedo (Elvidge et al., 2020)'

**Page 17, line 377:** include *'during such conditions'* after '*…key role in melt occurrence*'.

Thanks, done.

**Page 18, line 379:** Did you look into the melt immediately after foehn events? 6-12 hours after the foehn event for example? It could provide a nice extra result.

We have had a look at the diurnal cycle in energy fluxes and melt rates on the days before and after days with foehn, as from Figure 9 is visible that some of the days with melt occur after foehn events. Looking at a few hours after a foehn event will be biased by the diurnal cycle as melt solely occurs when solar radiation is peaking. We did not include these diurnal cycle plots on the days after foehn since it was hard to draw conclusions from them based on this short study period where also days after foehn overlap with days before the next foehn event. This idea that large melt rates might occur after foehn events will be considered in follow-up study using a longer term foehn-SEB record.

**Page 18, line 382:** In the brackets for the figure citation, include that the reader should look for the grey line on 9a, as it took me a while to see it amongst the other colours and bars.

Thanks, we have added this.

**Page 19, line 397:** include '*in which*' after '*68% of the hours*'

Thanks, done.

**Page 19, line 398:** change '*is peaking*' to '*peaks*'

Thanks, done.

**Page 19, figure 10:** Consider changing the colourbar to one that isn't rainbow, as this can be difficult for those with colourblindness. Interesting way to represent wind direction – I like it! Same for figure C1.

It was difficult to find a colourblind-proof cyclic colormap, therefore we changed it into wind vectors as suggested by reviewer 2. We adjusted the colour palette of the line graphs in the manuscript as well to make them more colourblind-friendly.

**Page 20, line 410:** is this steady south-easterly a product of your foehn algorithm though?

This can indeed be partly caused by the foehn algorithm. However, we found that running the Speirs algorithm without the wind-direction criterion gives very similar results, suggesting that this constant wind-direction is typical for foehn conditions and a clear deviation in summer from normal conditions where a diurnal wind regime occurs.

**Page 20: first paragraph of section 5:** Some of this novelty should be clearer in the introduction.

We have added the following sentence in line with your previous suggestion around the radiation observations in the Introduction: 'This is the first SEB study in the MDV to use a glacier based 4-component radiometer, which allowed for calibration of the simulated surface temperatures and increases accuracy of the simulated energy fluxes.'

**Page 21, line 453:** You also had quite a short duration of data, so could not look at the interannual variability or be sure that this 15% value is characteristic for your region either.

We agree with this, we can only say it is within the inter-annual variability of the northern MDV foehn. We clarified this in the manuscript by adding: 'The average foehn occurrence found for the 14 month study period at Joyce Glacier might not reflect the climatological mean, but it lies within the large inter-annual variability of the northern MDV (Speirs et al. 2013)'.

**Page 21, line 460:** The relatively coarse vertical resolution of AMPS (44 levels) may also not allow simulation of conditions closer to the surface. Many regional models now have 60-70 vertical levels.

That's right. Some recent studies use even higher resolution models to study foehn events (e.g. Umek et al 2022) and show that vertical resolution improves the simulation of cold-pool and foehn interaction and therefore the foehn signature at the surface. We realize that AMPS might not fully represent the conditions close to the surface and therefore we use AWS observations for the near-surface conditions and use AMPS solely to study the larger-scale atmospheric structure during foehn. We have added a statement around the vertical resolution in Section 3.1: 'Secondly, an accurate representation of near-surface conditions might be limited by the vertical resolution of the AMPS forecast'.

---

## Author Comment (AC2)

**Comments on reviewer #2**, the original review is given in black, and author comments in **blue.**

**Review of Hofsteenge et al. (2022): The surface energy balance during foehn events at Joyce Glacier, McMurdo Dry Valleys, Antarctica**

**Overall**
This study investigates the foehn and surface energy balance (SEB) over the Joyce Glacier, where receives less attention in the previous research. This paper use SEB model simulations that driven by the AWS observation and AMPS forecast outputs. Authors suggest that SWnet is the dominant driver during the daytime with a continuous contribution from sensible heat flux. Surface sublimation offsets the positive SEB, and reduced albedo further enhances the melting. This paper also highlights the important of solar radiation penetration for the accurate estimation of SEB. This paper includes several novel findings, and the scientific part is solid. A few results will be more convincing with more data or better explanations. The Discussion section might need some reorganizations. Thus, I would suggest a **minor revision** for this paper.

We would like to thank you for taking the time to review this manuscript. We are happy to read your comments and to incorporate the suggestions you made to improve the paper. We describe below in more detail which changes have been made to the manuscript.

**General**
**A. Structure:**
The conclusion and discussion part includes lots of interesting and novel findings. I think the reader will appreciate that if it is better organized. Authors first talked about SEB, then back to foehn (which contributes to the increase SWD via foehn clearance, SH via turbulence/downslope wind on the leeside, etc). The comparison between Joyce Glacier and AP is great. However, the connection between each topic is kind of loose.

We agreed that the discussion and conclusion part could be structured better. We have in the revised text chosen to separate it in three sections, highlighting first (Section 5.1) the differences in foehn mechanism in the complex MDV valleys, then (Section 5.2) the steps that have been made in simulating the energy balance of Joyce in relation to previous SEB studies in the MDV, and closing (Section 5.3) with the role of foehn on the SEB & melt of Joyce glacier in comparison with studies for the AP. We hope that this will guide the reader better and helps to highlight the main findings of the paper.

**B. Data:**
AMPS provides reliable forecast data for operational use. However, it has difficulties simulating the cloud conditions partially due to the microphysical scheme it is using (WSM5). This can lead to cold bias on the surface and compromise its ability to describe the SEB. Has author evaluated AMPS data with reanalysis data like ERA5 (31km) or ERA5 Land (9km)? It is not a serious problem, since this paper mainly analyzes SEB based on the SEB model. The surface variable (2mT, 10m Wind) needs a brief evaluation observation or reanalysis data. Also, can author explain why using a 12-h spin up instead of 24-h (more common)?

Thanks for your comments. As you mention, AMPS output is indeed not used for SEB simulations, but only for evaluation of the foehn detection methods and to describe the larger-scale meteorological setting during foehn events. At the start of section 3.1, AMPS output is compared to local observations. However, we mainly discuss differences in wind direction there and we have added a brief evaluation of Tsfc, RH, and 10m wind against observations in that paragraph, as follows: 'The AMPS 2m temperature shows a strong correlation with observed 2m temperature (0.97 using hourly data and 0.86 for hourly anomalies from the monthly mean to remove the impact of the annual cycle). The 2m temperature has a mean bias of -1C, with a larger negative bias in summer months (down to -3C) and a smaller bias in winter. A lower correlation is found for AMPS relative

humidity (0.45), with a mean bias of -5.7\%. AMPS 10m wind speed has a correlation coefficient of 0.5 with observations and mean bias of -1.2 m/s.'

Regarding the 12-h spinup for AMPS. Since the full AMPS output is not available anymore in an online archive, we have used a thinned AMPS product which was saved as the 12-24h forecast hours. We have in the revised text cited a paper (Seefeldt and Cassano, 2012) using AMPS in the same way who provided us with these AMPS products.

The cloud condition is important to back up some conclusions in this paper. I would suggest authors include cloud observation from satellite or simulation from reanalysis data (e.g., MODIS, ERA5, CERES, etc.) to better explain the SEB conditions (e.g., Ln 366).

We believe that one of the strengths of this study is that this is the first SEB study in the MDV that uses 4-component radiometer observations, as was also pointed out as a strength by reviewer 1. The high-quality data of incoming longwave radiation gives us the opportunity to calculate the equivalent longwave cloudiness. Since Lin is used for the SEB model as well, it gives a consistent way to analyse the impact of clouds on the radiation and ablation terms. We believe that this cloud estimation is the most accurate measure of cloudiness to represent the SEB at the AWS at Joyce. We decided not to include satellite and reanalysis data in this study as they are less likely to accurately represent the conditions at Joyce glacier, while they might be of value in studies focussing on the spatial impacts of clouds on the SEB.

**Minor:**
1. Echoing another reviewer's suggestion. Need more labels for the local topographic features in Fig. 1. Plus, the current labels are also hard to see.

Thank you for your comment, we have adjusted Figure 1 to clarify the valleys/topographic features that are mentioned throughout the paper. We hope this gives the reader more guidance.

2. Terms like foehn warming, isentropic drawdown, internal melting might need some brief description.

Thank you for pointing this out, we added a line explaining foehn the first time its mentioned: 'Foehn winds typically occur through topographic modification of flow in the lee of mountain barriers, resulting in strong and warm winds'. Isentropic drawdown is explained in Section 2.4.3 the first time it is mentioned, as follows: 'Isentropic drawdown is the drawdown of the potentially warmer and drier air from higher elevations when airflow is forced over a mountain range and down the leeward side'. We explained internal melt shortly as follows: 'Subsurface melt in the snow or ice pack (hereafter called internal melt) … '.

3. I would suggestion authors use forecast output/forecast for AMPS, and simulation for SEB model outputs.

Thank you, we agree the use of simulation was confusing and we adopted your suggestion to use 'simulation' when referring to SEB model outputs and 'forecast' to refer to AMSP output.

4. Are you able to differentiate the foehn wind with katabatic wind in your study?

The wind events that we detect as foehn events are unlikely to be katabatics as they are associated with cyclone activity and synoptic flow that leads to foehn events rather than air originating from the Antarctic plateau. We have checked the individual pressure fields that compose the composite plot of sea level pressure anomaly during foehn and see that these wind events consistently occur with cyclones over the ross sea and an associated synoptic flow towards the MDV. We have adjusted the structure of the Discussion & Conclusion section and have included more discussion around the katabatic vs foehn origin of these winds here to clarify this point.

Ln 300: Have authors checked the vertical wind profile or calculated the Fr number to confirm the low-level blocking? How strong the blocking is?

We do not have the vertical wind component in the AMPS product that is used in this study, therefore no wind vectors are included in the cross-section plots. We have checked the horizontal winds in these profiles, and they showed reduced windspeeds at the low levels on the windward side of the Royal Society Range, indicating blocking. We have calculated the Froude number from the AMPS fields, which confirms that the low-level flow is blocked. We have extended the description around the flow blocking in the paper with the description of the Froude number, as follows: 'To confirm the blocking flow regime, the Froude number is calculated as $Fr = U/Nh$, with h being the height of the Royal Society Range (~ 2.5 km), U the average windspeed of the upwind flow until mountain height and N the Brunt Väisälä frequency. We find an average Froude number of 0.2 during foehn and it never exceeds 0.7, indicating that the low-level flow is blocked and the flow cannot ascend over the Royal Society Range, but rather diverts around it.'

Ln 385: Some satellite or reanalysis data can be used here to provide some cloud information, such as cloud cover, phase, height, and thickness. Also, I am curious if there is any cloud formation, where is the moisture source?

We refer to our response to your comment around Ln 366 for our decision to not use satellite or reanalysis data. Figure C1 in the Appendix shows the average diurnal conditions during overcast and clear sky days and indicates that the moisture source is likely from McMurdo sound, evident from the steady sea breeze on overcast days. We do think it is interesting to study further the role of clouds and cloud formation in the MDV, since glacial melt is so strongly radiation driven. However, including this is outside the scope of this current paper.

Ln 425: Does the solar radiation penetration associated with the duration of the clear-sky conditions? Will the accurate description of the changing surface (wet snow, melting pond, refreeze) benefit the estimation of solar radiation penetration?

The solar radiation penetration in the model is wavelength dependent, and parameterized with the zrad parameter, which in practice regulates the fraction of solar radiation that is considered for surface temperature calculation and the part that goes into the subsurface. The solar penetration is dependent as well on the grain radius, which is different for snow and ice. It is therefore mainly important to represent in the model well whether the surface is covered by snow or an ice surface, which is dependent on the precipitation forcing. We have checked if the snow/ice cover is represented accurately by comparing it to the observed albedo and see that the main snow events are captured well in the simulation. We included some more information on the difference between snow/ice cover and the radius used for snow and ice in the methods section, as follows: 'Solar radiation can penetrate to considerable depths when the surface is ice and is reduced when snow-covered. A snow grain radius of 0.2 mm and an ice grain radius of 2.8 mm (based on measurements at Taylor Glacier, Hoffman et al. 2018) are used for the solar radiation penetration routine.'. The representation of liquid water in the snowpack or at the surface is not evaluated as we don't have observations that allow us to do so.

Ln 475: I think whether the downslope wind is able to reach the ice surface is highly associated with the mountain waves on the leeside, stationary orographic gravity, hydraulic jump, or breaking waves? This is not only associated with the magnitude of foehn, but also the detailed mechanism.

Thank you, we think you are right that the strength of foehn warming/drying at the surface is highly linked to the ability of the foehn air to reach the surface, which is in turn linked to the wave activity and occurrence of hydraulic jumps. We have elaborated on this as follows: 'In addition, the strength

of the foehn warming and drying at the surface relates to the ability of the foehn air to reach the ice surface, which is likely associated with lee-side wave activity and occurrence of hydraulic jumps.'.

Fig. 5 the wind vector reference is hard to see. Which AMPS domain is used here (02?) at what resolution (15km)? what is the baseline for the anomaly? I would suggest including those info in the caption.

We have clarified the caption of Figure 5, explaining better how the anomaly is calculated and including the resolution and domain of the AMPS fields, as follows: 'The foehn anomaly is calculated as the mean of the AMPS fields during foehn hours at Joyce Glacier minus the mean during hours without foehn.'. The wind vector reference is not placed outside the figure to make it easier to see.

Fig 10&C1. Just a suggestion. Will that be better to use wind barbs to indicate the direction instead of different colors?

In line with the comment of Reviewer 1 that the colormap is not colourblind friendly, we have indicated the wind direction with wind vectors now.

---

## Author Response (AR2)

**Response to Editor comments**

Dear Authors,

Thank you for your efforts and thoughtful revisions in response to reviewer comments. I am pleased to accept your interesting manuscript for publication in TC subject to addressing a few, largely technical last points:

1. Reviewer 1 asked "Does the choice of a different threshold affect when foehn is detected, and how did you settle on the specific directions you use?" I think this is a valid point and would be happy to see a more detailed answer to this question as well as a clearer justification for the specific ranges added to the methods section.

The wind direction criteria were chosen to make sure that the wind direction does not allow for false detection of foehn winds, for example through detecting down-glacier winds or the sea-breeze. We have clarified this as follows:

'A different wind direction criterion is used on AWS data compared to AMPS forecast, since the weather station was located close to the valley side and AMPS does not capture the topographic modification of the winds very well (Sect. 3.1). The range in wind-direction is chosen so that it excludes wind-directions that are expected during sea-breezes and down-glacial winds'

2. L118: Is the first part of this sentence correct? Measurements are not my speciality but S_in and S_out are usually measured (here presumably with the 4-component radiometer) while albedo is calculated.

We see this sentence is confusing. Indeed, albedo is calculated from S_in and S_out, however the model calculates the net shortwave radiation that is used in the energy balance from the forced S_out and accumulated albedo, to reduce errors introduced by errors in the S_in observations. We have rephrased this sentence to: 'S_net (S_in (1- a)) is calculated from observed S_out and accumulated albedo as suggested by Van den Broeke et al. (2004).'

3. L153: Please provide an indication of the atmospheric depth covered by the first 15 model levels in AMPS.

We have included this as follows: 'AMPS is run on 44 model levels, and this study uses AMPS products that were reduced to the first 15 model levels (covering approximately the first 1.5 km above the surface) and 6 fixed pressure levels.'

4. L302: I suggest adding "(not shown)" since this specific result is not visible to the reader.

We have included this in the revised manuscript.

5. Figure 10: I like the idea to use vectors but have trouble interpreting them. What does the length correspond to, if the origin indicates 10*windspeed? Can you please provide a reference vector as well as some indication in the caption which directions are up/down glacier for ease of reading?

We have added a reference vector next to the legend in both Figure 10 and C1 and have added in the caption the following: 'Arrows indicate wind-direction, with typically a north-easterly sea-breeze, south-westerly down-glacier wind and south-easterly foehn wind.'.

Thank you and best regards,
Emily Collier